# Disrupting Deep Uncertainty Estimation Without Harming Accuracy

**Ido Galil**
Technion
idogalil.ig@gmail.com

**Ran El-Yaniv**
Technion,   Deci.AI
rani@cs.technion.ac.il

## Abstract

Deep neural networks (DNNs) have proven to be powerful predictors and are widely used for various tasks. Credible uncertainty estimation of their predictions, however, is crucial for their deployment in many risk-sensitive applications. In this paper we present a novel and simple attack, which unlike adversarial attacks, does not cause incorrect predictions but instead cripples the network's capacity for uncertainty estimation. The result is that after the attack, the DNN is more confident of its incorrect predictions than about its correct ones **without** having its accuracy reduced. We present two versions of the attack. The first scenario focuses on a black-box regime (where the attacker has no knowledge of the target network) and the second scenario attacks a white-box setting. The proposed attack is only required to be of minuscule magnitude for its perturbations to cause severe uncertainty estimation damage, with larger magnitudes resulting in completely unusable uncertainty estimations. We demonstrate successful attacks on three of the most popular uncertainty estimation methods: the vanilla softmax score, Deep Ensembles and MC-Dropout. Additionally, we show an attack on SelectiveNet, the selective classification architecture. We test the proposed attack on several contemporary architectures such as MobileNetV2 and EfficientNetB0, all trained to classify ImageNet.

## 1   Introduction

Deep neural networks (DNNs) show great and improving performance in a wide variety of application domains including computer vision and natural language processing. Successful deployment of these models, however, is critically dependent on providing effective *uncertainty estimation* for their predictions, or employing some kind of *selective prediction* [8].

In the context of classification, practical and well-known uncertainty estimation techniques include: (1) the classification prediction's *softmax score* [3, 4], which quantifies the embedding margin between an instance to the decision boundary; (2) *MC-Dropout* [7], which is argued to proxy *Bayesian networks* [22, 20, 23]; (3) *Deep Ensembles* [18], which have shown state-of-the-art results in various estimation settings; and (4) *SelectiveNet* [9], which learns to estimate uncertainty in a way that produces optimal risk for a specific desired coverage.

In this paper we show that all these well-known uncertainty estimation techniques are vulnerable to a new and different type of attack that can completely destroy their usefulness and that works in both *black-box* (where the attacker can only query the attacked model for predicted labels and has no knowledge of the model itself) and *white-box* (the attacker has complete knowledge about the attacked model) settings. While standard adversarial attacks target model accuracy, the proposed attack is designed to *keep accuracy performance intact* and refrains from changing the original classification predictions made by the attacked model. We call our method ACE: *Attack on Confidence Estimation*. To demonstrate the relevance of our findings, we test ACE on modern architectures, such

35th Conference on Neural Information Processing Systems (NeurIPS 2021).

as MobileNetV2 [25], EfficientNet-B0 [28], as well as on standard baselines such as ResNet50 [13], DenseNet161 [14] and VGG16 [26]. Our attacks are focused on the task of ImageNet classification [5]. Figure 1 conceptually illustrates how ACE works. Consider a classifier for cats vs. dogs that

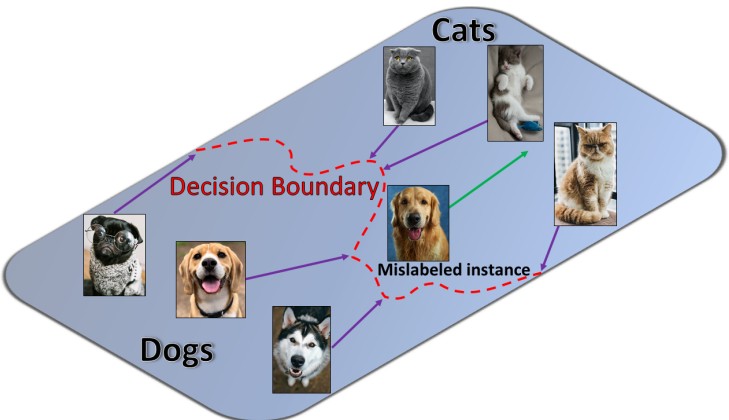

Figure 1: The intuition driving ACE on a classifier of cats and dogs, when the uncertainty measure used is softmax. Correct predictions are moved closer to the decision boundary without crossing it, thus increasing their uncertainty (purple arrows), and incorrect predictions are moved further away from the decision boundary to decrease their uncertainty (green arrow).

uses its prediction's softmax score as its uncertainty estimation measurement. An end user asks the model to classify several images, and output only the ones in which it has the most confidence. Since softmax quantifies the margin from an instance to the decision boundary, we visualize it on a 2D plane where the instances' distance to the decision boundary reflect their softmax score. In the example shown in Figure 1, the classifier was mistaken about one image of a dog, classifying it as a cat, but fortunately its confidence in this prediction is the lowest among its predictions. A malicious attacker targeting the images in which the model has the most confidence would want to increase the confidence in the mislabeled instance by pushing it away from the decision boundary, and decrease the confidence in the correctly labeled instances by pushing them closer to the decision boundary.

Our attack, however, can be accomplished without harming the model's accuracy, and thus avoid raising suspicion about a possible attack. Even if the mislabeled dog is assigned more confidence than only a single correctly predicted image, ACE will undermine the model's uncertainty estimation performance, since based on its estimation measurements, it cannot offer its newly least-confident (and correct) instance without offering the now more-confident mislabeled dog. It will either have to exclude both from the predictions it returns, reducing its coverage and its usefulness to the user, or include both and return predictions with mistakes, increasing its risk.

ACE, unlike adversarial attacks, creates damage even with small magnitudes of perturbations, befitting even an attacker with very limited resources (see Section 5). This benefit is inherent in our goal, which is different from standard adversarial attacks: adversarial attacks seek to move the attacked instance across the model's decision boundary, and thus require the magnitude of the perturbations it causes to be sufficiently large to allow this. Our objective is to simply move the correctly and incorrectly predicted instances, relative to one another within the boundaries, which requires a less powerful "nudge". Figure 2 exemplifies such an attack on EfficientNet, showing how the model would be more confident of binoculars being a tank than a correctly-labeled tank. Figure 6 shows the *Risk-Coverage* (RC) curve (see Section 3 for an explanation of RC curves) for EfficientNet under our attack, and that for the strongest attack tested (in terms of perturbation magnitude, which is still a very small amount), approximately 20% of the model's most confident predictions were wrong.

Uncertainty attacks are in general conceptually different than adversarial attacks. Consider a scenario of a debt landing company using machine learning to decide whether to approve or disapprove loans to its clients. An interesting example is: lendbuzz.com. Such a company operates by granting loans to highly probable non-defaulting applicants achieving the highest confidence according to their predictive model. The choice of rejection threshold must depend on uncertainty estimation (in accordance with the company's risk tolerance). A malicious attacker might attempt to influence this company to grant loans to the maximal amount of defaulting applicants to increase its odds to go

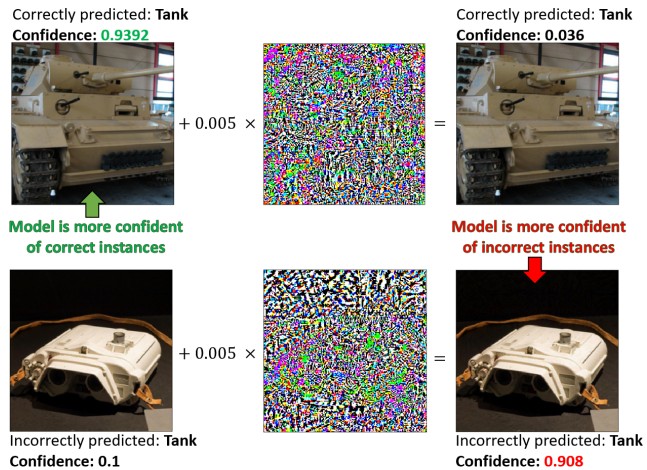

Figure 2: Demonstration of attacking EfficientNet with softmax as the uncertainty estimation method. Left: Two images being predicted to be tanks. The top one is correctly labeled and the model has high confidence in its label. The bottom image is of binoculars incorrectly labeled as a tank, with the model having low confidence in its prediction. Right: After adding perturbations to the original images, the model has lower confidence in its (correct) tank label than in the wrongly labeled binoculars.

bankrupt. Aware of this risk, the company tries to defend itself in various ways. Two obvious options are: (1) Monitor their model's accuracy. if it drops significantly it might point to an adversarial attack. Since normal adversarial attacks harms accuracy significantly, it will alarm such monitors. ACE does not harm accuracy, and therefore will not. (2) Subscribe to AI firewall services like those given by professional parties such as robustintelligence.com. Since standard adversarial attacks generate instances that cross the decision boundary, they require a relatively large epsilon that is easier to detect relative to those required by ACE. Even if the standard adversarial attack bypassed detection, the crossing adversarial instance they generated may have a very high uncertainty estimate due to its proximity to the decision boundary.

While a similar attack was already considered in the context of OOD detection with Dirichlet-based uncertainty models [15], the present paper is the first to offer an algorithm to attack only the uncertainty estimation of any model employing any of the most common uncertainty estimation techniques in both black-box and white-box settings.

## 2    Related work

The most common and well-known approach to estimate uncertainty in DNNs for classification is using softmax at the last layer. Its values correspond to the distance from the decision boundary, with high values indicating high confidence in the prediction. The first ones to suggest this approach using DNNs were [3, 4].

Lakshminarayanan et al. [18] showed that an ensemble consisting of several trained DNN models gives reliable uncertainty estimates. For classification tasks, the estimates are obtained by calculating a mean softmax score over the ensemble. This approach was shown to outperform the softmax response on CIFAR-10, CIFAR-100, SVHN and ImageNet [11].

Motivated by Bayesian reasoning, Gal and Ghahramani [7] proposed *Monte-Carlo dropout* (MC-Dropout) as a way of estimating uncertainty in DNNs. At test time, for regression tasks this technique estimates uncertainty using variance statistics on several dropout-enabled forward passes. In classification, the mean softmax score of these passes is calculated, and then a predictive entropy score is calculated to be used as the final uncertainty estimate.

Geifman and El-Yaniv [9] introduced SelectiveNet, an architecture with a suitable loss function and optimization procedure that trains to minimize the risk for a specific required coverage. The SelectiveNet architecture can utilize any DNN as its backbone, and constructs three prediction heads on top of it – one of which is used solely for training, and the other two for prediction and rejection.

SelectiveNet incorporates a soft "reject option" into its training process, allowing it to invest all its resources to achieve an optimal risk for its specific coverage objective. It was shown to achieve state-of-the-art results on several datasets for deep *selective classification* across many different coverage rates.

The *area under the risk-coverage curve* (AURC) metric was defined in [10] for quantifying the quality of uncertainty scores. This metric is motivated from a selective classification perspective: given a model, and using any uncertainty score for that model, an RC curve can be generated to indicate what the resulting risk ($y$-axis) would be for any percentage of coverage ($x$-axis). AURC is then defined to be the area under this curve. More details and an example are provided in Section 3.

Turning to *adversarial attacks*, one of the first *adversarial examples* was presented by Szegedy et al. [27]. They showed how small, barely perceptible perturbations of the input to a CNN, crafted by using the box-constrained L-BFGS (Limited-Memory Broyden-Fletcher-Goldfarb-Shanno) algorithm, could alter the DNN's classification. Goodfellow et al. [12] developed a more efficient way to craft adversarial examples, called the *fast gradient sign method* (FGSM), requiring only a single backward pass of the network. Formally, with $\epsilon$ being a fixed magnitude for the image perturbation, $y$ being the true label and a given $loss$ function, the FGSM adversarial example is simply $\tilde{x} = x + \epsilon \cdot sign(\nabla_x loss(f(x), y))$.

Liu et al. [19] demonstrated that crafting transferable adversarial examples (examples fooling multiple models, possibly of different architectures) for black-box attacks is especially challenging when targeting specific labels. The authors showed that using an ensemble results in highly transferable adversarial examples. The knowledge required for this method to create non-targeted attacks is only the ground-truth label and the attacked model's predicted label for targeted attacks.

## 3 Problem setup

Let $\mathcal{X}$ be the image space and $\mathcal{Y}$ be the label space. Let $P(\mathcal{X}, \mathcal{Y})$ be an unknown distribution over $\mathcal{X} \times \mathcal{Y}$. A model $f$ is a prediction function $f : \mathcal{X} \rightarrow \mathcal{Y}$, and its predicted label for an image $x$ is denoted by $\hat{y}_f(x)$. The model's *true* risk w.r.t. $P$ is $R(f|P) = E_{P(\mathcal{X}, \mathcal{Y})}[\ell(f(x), y)]$, where $\ell : \mathcal{Y} \times \mathcal{Y} \rightarrow \mathbb{R}^+$ is a given loss function, for example, 0/1 loss for classification. Given a labeled training set $S_m = \{(x_i, y_i)\}_{i=1}^m \subseteq (\mathcal{X} \times \mathcal{Y})$, sampled i.i.d. from $P(\mathcal{X}, \mathcal{Y})$, the *empirical risk* of the model $f$ is $\hat{r}(f|S_m) \triangleq \frac{1}{m} \sum_{i=1}^m \ell(f(x_i), y_i)$.

For a given model $f$, following [10], we define a *confidence score* function $\kappa(x, \hat{y}|f)$, where $x \in \mathcal{X}$, and $\hat{y} \in \mathcal{Y}$ is the model's prediction for $x$. The function $\kappa$ should quantify confidence in the prediction of $\hat{y}$ for the input $x$, based on signals from the model $f$. This function should induce a partial order over points in $\mathcal{X}$, and is not required to distinguish between points with the same score. For example, for any classification model $f$ with softmax at its last layer, the vanilla confidence score is its softmax response values: $\kappa(x, \hat{y}|f) \triangleq f(x)_{\hat{y}}$. Note that we are not concerned with the values of the confidence function; moreover, they do not have to possess any probabilistic interpretation. They can be any value in $\mathbb{R}$. An optimal $\kappa$ for a given classification model $f$ and 0/1 loss should hold for every two instances with their ground truth, $(x_1, y_1) \sim P(\mathcal{X}, \mathcal{Y})$ and $(x_2, y_2) \sim P(\mathcal{X}, \mathcal{Y})$:

$$\kappa(x_1, \hat{y}_f(x_1)|f) \leq \kappa(x_2, \hat{y}_f(x_2)|f) \iff \mathbf{Pr}_P[\hat{y}_f(x_2) \neq y_2] \leq \mathbf{Pr}_P[\hat{y}_f(x_1) \neq y_1]$$

so that $\kappa$ truly is monotone with respect to the loss.

In practice, DNN end users often calibrate a threshold over the $\kappa$ used (whether it is a softmax one or an MC-dropout one, etc.) to get the desired coverage over the inference inputs exceeding this threshold, and give special treatment to the rest (such as consulting an expert or abstaining entirely from giving a prediction). A natural way to evaluate the performance of a $\kappa$ in such settings is from a *selective model* perspective. A *selective model $f$* [6, 2] uses a *selection function* $g : \mathcal{X} \rightarrow \{0, 1\}$ to serve as a binary selector for $f$, enabling it to abstain from giving predictions for certain inputs. $g$ can be defined by a threshold $\theta$ on the values of a $\kappa$ function such that $g_\theta(x|\kappa, f) = \mathbb{1}[\kappa(x, \hat{y}_f(x)|f) > \theta]$. The performance of a selective model is measured using coverage and risk, where *coverage*, defined as $\phi(f, g) = E_P[g(x)]$, is the probability mass of the non-rejected instances in $\mathcal{X}$. The *selective risk* of the selective model $(f, g)$ is defined as $R(f, g) \triangleq \frac{E_P[\ell(f(x), y)g(x)]}{\phi(f, g)}$. These quantities can be evaluated empirically over a finite labeled set $S_m$, with the *empirical coverage* defined as $\hat{\phi}(f, g|S_m) = \frac{1}{m} \sum_{i=1}^m g(x_i)$, and the *empirical*

*selective risk* defined as $\hat{r}(f, g|S_m) \triangleq \frac{\frac{1}{m} \sum_{i=1}^{m} \ell(f(x_i), y_i) g(x_i)}{\hat{\phi}(f, g|S_m)}$. The *risk-coverage curve* (RC curve)

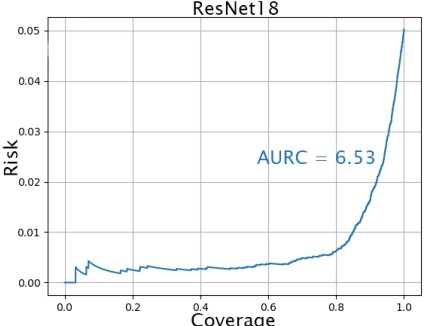

Figure 3: The RC curve of ResNet18 trained on CIFAR-10, measured on the test set. The risk is calculated using a 0/1 loss (meaning the model has about 95% accuracy for 1.0 coverage); the $\kappa$ used was softmax-response. The value of the risk at each point of coverage corresponds to the selective risk of the model when rejecting inputs that are not covered at that coverage slice. e.g., the selective risk for coverage 0.8 is about 0.5%, meaning that an end user setting a matching threshold would enjoy a model accuracy of 99.5% on the 80% of images the model would not reject.

is a curve showing the selective risk as a function of coverage, measured on some chosen test set. Figure 3 shows an example of such a curve. The RC curve could be used to evaluate a confidence score function $\kappa$ over all its possible thresholds (such that the threshold is determined to produce the matching coverage in the graph). A useful scalar to quantify $\kappa$'s performance is to consider all possible coverages, by calculating the area under its RC curve (*AURC*) [11], which is simply the mean selective risk.

## 4 How to attack uncertainty estimation

Attacking the uncertainty estimation "capacity" of a model is essentially sabotaging the partial order induced by its confidence score $\kappa$, for example, by forcing $\kappa$ to assign low confidence scores to correct predictions, and high confidence scores to incorrect predictions (an even more refined attack, discussed in Section 6, is to completely reverse the partial order, which could be useful when the loss used is not 0/1). An adversarial example that specifically targets uncertainty estimation can be crafted by finding a minimal perturbation $\epsilon$ for an input $x$, such that $\tilde{x} = x + \epsilon$ would cause $\kappa$ to produce a bad (in terms of selective risk) partial order on its inputs **without** changing the model's accuracy. The most *harmful* attack on $\kappa$ for a model $f$ should hold for every two adversarial instances with their ground truth:

$$\kappa(\tilde{x_1}, \hat{y}_f(\tilde{x_1})|f) \leq \kappa(\tilde{x_2}, \hat{y}_f(\tilde{x_2})|f) \iff \mathbf{Pr}_P[\hat{y}_f(\tilde{x_1}) \neq y_1] \leq \mathbf{Pr}_P[\hat{y}_f(\tilde{x_2}) \neq y_2].$$

Figure 5 shows such a hypothetical worse-case RC curve scenario, in which all of its incorrectly predicted instances are assigned more confidence than its correctly predicted ones. An end user setting a threshold to get a coverage of 0.22 over inputs would suffer a 100% error rate. An RC curve of our proposed attack showing similar actual damage can be observed in Figure 6.

The restriction of not changing the model's accuracy allows an attacker to only harm the uncertainty estimation without attracting suspicion by causing a higher error rate for the model. This constraint underscores the fact that it is not necessary to change the model's predictions in order to harm its capacity to estimate uncertainty. This restriction could be lifted based on the goal of the attack (if, for example, the attacker does not mind that the attacked model's accuracy is being reduced).

This type of attack also enjoys two benefits unique to it and not shared with common adversarial attacks: Firstly, the perturbations needed for this type of attack are inherently smaller than those needed for most adversarial attacks. Perturbations with too large magnitudes could cause the crafted example to cross the decision boundary and thus harm the model's accuracy (violating our restriction). Secondly, its success is not binary: The goal of most adversarial attacks is to push an input over the decision boundary. If the perturbation magnitude was insufficient to do that, the attack has failed. In

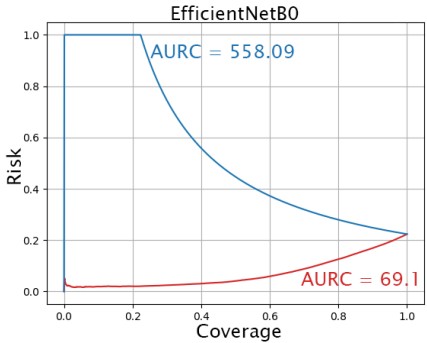

Figure 4: The blue curve shows the hypothetical worse possible RC curve of EfficientNetB0 trained on ImageNet, compared to its actual empirically measured RC curve (red curve).

our approach, the attack is partially successful even if it causes a single incorrect prediction to get a higher confidence score than a correct one, with the magnitude of the perturbations only contributing to the potency of the attack (creating a worse partial order and creating a larger confidence gap, numerically, between the correct and incorrect samples).

---

**Algorithm 1** Attack on Confidence Estimation

---

1: **function** ACE($f, \hat{f}, \kappa, x, y, \epsilon, \epsilon_{decay}, max\_iterations$)
2:     $\eta \leftarrow sign(\nabla_x \kappa(x, \hat{y}_f(x)|\hat{f}))$                     $\triangleright \kappa$ is calculated with $\hat{f}$ on the label predicted by $f$
3:     **for** $i < max\_iterations$ **do**
4:         **if** $\hat{y}_f(x) == y$ **then**                   $\triangleright$ f is correct, decrease confidence for $x$
5:             $\tilde{x} \leftarrow x - \epsilon \cdot \eta$
6:         **else**                             $\triangleright$ f is incorrect, increase confidence for $x$
7:             $\tilde{x} \leftarrow x + \epsilon \cdot \eta$
8:         **if** $\hat{y}_f(\tilde{x}) == \hat{y}_f(x)$ **then**
9:             **return** $\tilde{x}$              $\triangleright$ label is unchanged so accuracy will be unharmed
10:         **else**
11:             $\epsilon \leftarrow \epsilon \cdot \epsilon_{decay}$
12:     **return** $x$                      $\triangleright$ insufficient $max\_iterations$ & $\epsilon$ too big

Algorithm's arguments: f is the attacked model, $\hat{f}$ is the proxy for $f$, $\kappa$ is the confidence scoring function, $x$ is the input, y is the true label, $\epsilon$ is the initial magnitude of the perturbation, $\epsilon_{decay}$ is the rate at which $\epsilon$ decreases when the attack changes the input's prediction and $max\_iterations$ is the number of $\epsilon$ decrease steps.

---

Algorithm 1 summarizes the general template for our proposed attack, dubbed 'ACE' (Attack on Confidence Estimation). In black-box settings, the proxy $\hat{f}$ for the attacked model $f$ is an ensemble of models not consisting of $f$ itself (with $\hat{y}_f(x)$ requiring a query to $f$ to obtain its label for the input $x$), and in white-box settings, $\hat{f} = f$. The derivative of $\kappa$ w.r.t. $x$, in the case of the softmax score, is the derivative of the $y$ prediction component of $\hat{f}$ w.r.t. to the input $x$, where $y = y_f(x)$. We show in our white-box experiments that attacking the softmax score specifically is sometimes preferable, even if the $\kappa$ used is not the softmax score, which is why we target the softmax score in black-box settings. We speculate that this is due to the fact that softmax allows us a direct "handle" on the prediction's margin in cases where $\kappa$ does not (for example, in Section 5.4 we show that targeting the selector head of SelectiveNet that is used for rejection is less effective than targeting its softmax score).

While the scope of this paper is classification, future work could modify ACE for regression tasks, in which the uncertainty is often measured by the variance of several outputs made by the model (e.g., the variance of outputs provided by an ensemble of DNNs). A simple conversion of this algorithm could, for example, define any instance with a loss above some threshold (such as the median loss on some validation set) as an "incorrect prediction" and loss values below the threshold as "correct predictions". The $\kappa$ being attacked would be the variance of the outputs, causing an increase in

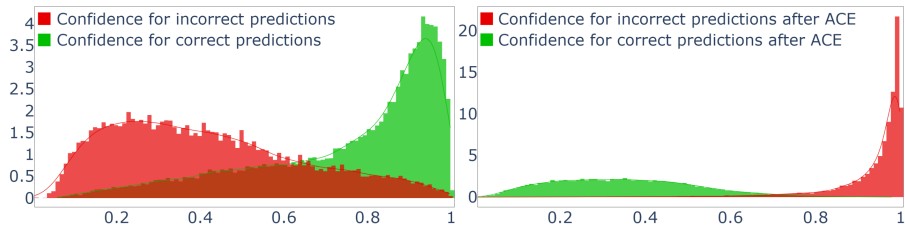

Figure 5: Left: histograms of an EfficientNet's confidence scores (with $\kappa$ being softmax-response) for its correct predictions and incorrect predictions. Right: histograms for the confidence scores after using ACE ($\epsilon = 0.005$, white-box) to attack EfficientNet's uncertainty estimation performance.

variance for the inputs for which the model had a low loss, and a decrease in variance for inputs for which the model had a high loss.

A weakness of ACE for conducting large scale attacks is that knowledge of the ground truths for the instances it attacks is required, which might be difficult to obtain. Some attackers, however, could have a good proxy of the ground truths, such as a different model capable of making accurate predictions. Alternatively, the attacker might not care at all about the ground truths and would like only to increase the confidence for certain types of labels and decrease the confidence for other types regardless of their truthfulness (i.e., causing a loan company to have a very high confidence for the label approving the granting of a loan and a very low confidence for the label disapproving loans, thus increasing the chance for it to go bankrupt).

## 5 Experiments

We evaluate the potency of ACE using several metrics: AURC ($\times 10^3$), *Negative Log-likelihood* (NLL) and *Brier score* [1] commonly used for uncertainty estimation in DNNs, and defined in Appendix A. We test several architectures on the ImageNet [5] validation set (which contains 50,000 images), implemented and pretrained by PyTorch [24], with the exception of EfficientNetB0, which was implemented and pretrained by [29]. We test ACE in black-box settings using three values for $\epsilon$: 0.0005, 0.005 and 0.05 (with RGB intensity values between 0 and 1). We follow [19] in using an ensemble to craft the adversarial examples, in our case consisting of ten ResNet50 models trained on ImageNet. For "easier" white-box settings we test ACE with even smaller $\epsilon$ values: 0.00005, 0.0005 and 0.005. To the best of our knowledge, some of our tested $\epsilon$ values were never used by standard adversarial attacks. Due to ACE refraining from changing the originally predicted labels and decreasing $\epsilon$ iteratively if needed, the effective $\epsilon$ values selected by ACE are smaller and noted in the results (as "Effective $\epsilon$"). In all of our experiments, we set the hyperparameters arbitrarily: $\epsilon_{decay} = 0.5, max\_iterations = 15$. We did not try to optimize these, and we expect the algorithm to reach better results with a smaller $\epsilon_{decay}$ if we also increase $max\_iterations$ (since it would find bigger $\epsilon$ values that could be used without changing the predictions).

### 5.1 Attacking softmax uncertainty

Table 1 shows the results of using ACE for different values of $\epsilon$ under black-box settings. Note that for larger values of $\epsilon$, the effective $\epsilon$ is about half the size, meaning that even fewer resources are sufficient for a very harmful attack. Under attack by ACE with $\epsilon = 0.05$, the AURC is $3.5 - 6.5$ times worse, and both the NLL and Brier score have worsened by about 100%, meaning that the uncertainty estimation of the model is devastated while the accuracy is left unharmed – just as ACE intended. The RC curves included in Appendix B.3 provide an important insight the metrics cannot: while on average, larger $\epsilon$ values harm uncertainty estimation more, sometimes for larger coverages, a smaller $\epsilon$ is more harmful. For example, the selective risk for VGG16 above coverage 0.6 is *higher* for $\epsilon = 0.005$ than it is for $\epsilon = 0.05$ (indicated by the purple curve being under the blue curve). This artifact may either be inherent to such black-box attacks, or it may be due to many attacked instances losing their "adversarial trajectory" by taking overly large steps, and could be fixed with an iterative version of ACE further discussed in Section 6. Figure 6 shows an RC curve for EfficientNetB0 with ACE under white-box settings (the legend presents the different values for $\epsilon$ ). Results for the even more potent white-box setting and its RC curves are included in Appendix B.1. In Appendix B.2 we

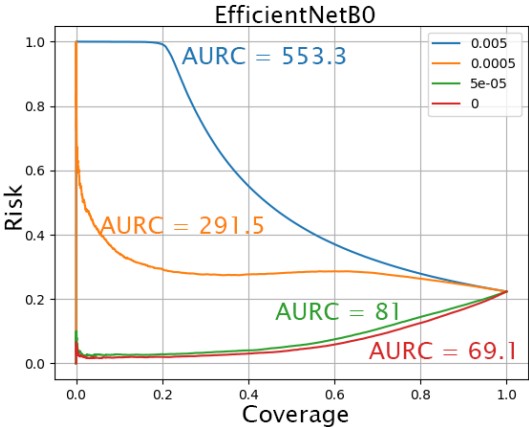

Figure 6: The RC curves of EfficientNet while under white-box attack by different values of $\epsilon$. The colored numbers next to each matching colored curve correspond to the area under the curve (AURC) $\times 1000$. For attacks with $\epsilon = 0.005$, an end user asking the model only for its 20% most confident predictions will get almost 100% wrong predictions.

evaluate ACE with an adversarially trained model. These results suggest that standard adversarial training does not provide significant robustness against ACE.

Table 1: Results of ACE under black-box settings on various architectures pretrained on ImageNet with softmax as the confidence score.

| Softmax score | $\epsilon$ | Effective $\epsilon$ | AURC | NLL | Brier Score | Accuracy |
|---|---|---|---|---|---|---|
| ResNet50 | 0 | 0 | 69.9 | 0.963 | 0.336 | 76.01 |
| | 0.0005 | 0.00046 | 141.9 | 1.308 | 0.474 | 76.01 |
| | 0.005 | 0.003353 | 436.9 | 2.615 | 0.652 | 76.01 |
| | 0.05 | 0.020622 | 456.6 | 2.494 | 0.656 | 76.01 |
| EfficientNetB0 | 0 | 0 | 69.1 | 0.958 | 0.322 | 77.67 |
| | 0.0005 | 0.000495 | 78.8 | 1 | 0.342 | 77.67 |
| | 0.005 | 0.004459 | 185.1 | 1.334 | 0.47 | 77.67 |
| | 0.05 | 0.029023 | 300.9 | 1.775 | 0.585 | 77.67 |
| Mobilenet V2 | 0 | 0 | 89.7 | 1.147 | 0.386 | 71.85 |
| | 0.0005 | 0.000493 | 104.3 | 1.217 | 0.416 | 71.85 |
| | 0.005 | 0.004276 | 236.9 | 1.745 | 0.578 | 71.85 |
| | 0.05 | 0.025108 | 317.8 | 2.148 | 0.628 | 71.85 |
| DenseNet161 | 0 | 0 | 66.8 | 0.945 | 0.326 | 77.15 |
| | 0.0005 | 0.000493 | 81 | 1.022 | 0.358 | 77.15 |
| | 0.005 | 0.00427 | 209.8 | 1.577 | 0.508 | 77.15 |
| | 0.05 | 0.026584 | 359.7 | 2.1 | 0.585 | 77.15 |
| VGG16 | 0 | 0 | 80.5 | 1.065 | 0.366 | 73.48 |
| | 0.0005 | 0.000493 | 93.7 | 1.135 | 0.395 | 73.48 |
| | 0.005 | 0.004296 | 207.8 | 1.634 | 0.544 | 73.48 |
| | 0.05 | 0.026088 | 298.2 | 1.985 | 0.602 | 73.48 |

## 5.2 Attacking deep ensemble uncertainty

Table 2: Results of ACE under black-box settings on an ensemble of size 5 consisting of ResNet50 models trained on ImageNet.

| ResNet50 Ensemble | $\epsilon$ | Effective $\epsilon$ | AURC | NLL | Brier score | Accuracy |
|---|---|---|---|---|---|---|
| Foreign Proxy | 0 | 0 | 63.8 | 0.883 | 0.317 | 77.61 |
| | 0.0005 | 0.00049 | 74.6 | 0.935 | 0.342 | 77.61 |
| | 0.005 | 0.00432 | 182.1 | 1.356 | 0.496 | 77.61 |
| | 0.05 | 0.02865 | 290.8 | 1.836 | 0.594 | 77.61 |
| ResNet50 Proxy | 0 | 0 | 63.8 | 0.883 | 0.317 | 77.61 |
| | 0.0005 | 0.00049 | 82.7 | 0.966 | 0.357 | 77.61 |
| | 0.005 | 0.00396 | 279.1 | 1.582 | 0.563 | 77.61 |
| | 0.05 | 0.02342 | 381 | 2 | 0.633 | 77.61 |

Table 2 shows the results of using ACE under black-box settings. We evaluate two different kinds of ensemble proxies to attack an ensemble of ResNet50: (1) an ensemble proxy of foreign architectures

(EfficientNet, MobileNetV2 and VGG), (2) an ensemble proxy consisting of ResNet50 models matching the victim models. While ACE harms the ensemble in both scenarios, it is clearly stronger when using a matching ensemble. In the resulting RC curve, which is included in Appendix C.2, we can make the same observation we did when attacking softmax: for large coverages, a smaller $\epsilon$ causes greater damage to the selective risk (using a ResNet50 proxy for example, the selective risk for any coverage above 0.45 is slightly higher for $\epsilon = 0.005$ than it is for $\epsilon = 0.05$). For white-box settings, we test ACE using ensembles of different sizes consisting of ResNet50 models trained on ImageNet. Appendix C.1 shows the results of these experiments, from which we observe that for $\epsilon = 0.005$, the AURC degrades about eightfold. An interesting observation is that the bigger the ensemble, the more resilient it is to ACE, and even the smallest ensemble is more resilient than a single ResNet50 model (in both black-box and white-box settings). Note that since the scope of this paper does not include explicit attempts to enhance these techniques' resiliency to our attack, we have not used adversarial training for the ensemble as was suggested by [18] (which was found to improve uncertainty estimation). We did not want to give a perceived advantage to Deep Ensembles over other techniques in this respect, and as the results show, Deep Ensembles are more resilient even without adversarial training.

## 5.3 Attacking Monte Carlo dropout

Table 3: Results for MC-Dropout of ACE under black-box settings on models pretrained on ImageNet with predictive entropy over several dropout-enabled passes producing the confidence score.

| MC-Dropout | $\epsilon$ | Effective $\epsilon$ | AURC | NLL | Brier Score | Accuracy |
|---|---|---|---|---|---|---|
| MobileNet V2 30 passes | 0 | 0 | 94.5 | 1.143 | 0.386 | 71.79 |
| | 0.0005 | 0.00049 | 104.4 | 1.207 | 0.413 | 71.79 |
| | 0.005 | 0.00427 | 212.6 | 1.71 | 0.575 | 71.84 |
| | 0.05 | 0.02514 | 275.2 | 2.145 | 0.634 | 71.79 |
| MobileNet V2 10 passes | 0 | 0 | 95.1 | 1.148 | 0.387 | 71.58 |
| | 0.0005 | 0.00048 | 104.9 | 1.21 | 0.413 | 71.72 |
| | 0.005 | 0.00427 | 213.2 | 1.713 | 0.573 | 71.7 |
| | 0.05 | 0.02514 | 275.3 | 2.146 | 0.631 | 71.61 |
| VGG16 30 passes | 0 | 0 | 86.1 | 1.056 | 0.366 | 73.22 |
| | 0.0005 | 0.00049 | 95 | 1.117 | 0.392 | 73.35 |
| | 0.005 | 0.0043 | 188.9 | 1.573 | 0.541 | 73.34 |
| | 0.05 | 0.02611 | 261.1 | 1.978 | 0.613 | 73.28 |
| VGG16 10 passes | 0 | 0 | 86.5 | 1.066 | 0.368 | 73.18 |
| | 0.0005 | 0.00048 | 95.9 | 1.123 | 0.392 | 73.13 |
| | 0.005 | 0.00429 | 189 | 1.578 | 0.537 | 73.08 |
| | 0.05 | 0.02608 | 261.5 | 1.984 | 0.609 | 73.04 |

We test ACE on MC-Dropout using predictive entropy as its confidence score. We first experiment with ACE under white-box settings and test two methods of attack: an *indirect method*, in which ACE targets the softmax score of a single dropout-disabled forward pass, and a *direct method*, in which ACE targets the predictive entropy produced from $N$ dropout-enabled forward passes. In both methods, the model's confidence in the prediction is calculated by predictive entropy over $N$ forward passes with dropout enabled. From the results presented in Appendix D.1, we can observe that while for small $\epsilon$ values the direct method is slightly more harmful, for bigger values the indirect softmax method is significantly more harmful and on par with results of attacking those models when their $\kappa$ is softmax instead of being calculated by MC-Dropout. These results lead us to conclude that for most purposes, an indirect softmax attack could be more harmful than a direct attack on the predictive entropy used by MC-Dropout. Using this knowledge, we attack MC-Dropout in black-box settings identically to how we attacked softmax: by using an ensemble as a proxy and attacking its softmax score, without using dropout at all. Observing Table 3 for the results, we notice that for black-box settings, MC-Dropout seems somewhat more resilient compared to using softmax as $\kappa$ for these same models. The number of forward passes made by the models do not seem to affect ACE's effectiveness. The RC curves for the black-box settings are provided in Appendix D.2.

## 5.4 Attacking SelectiveNet

Finally, we test ACE under white-box settings attacking SelectiveNet using two different architectures for its backbone, namely ResNet18 and VGG16, both trained on CIFAR-10 [16]. The confidence score is given by the selector head of SelectiveNet, which was trained to be used for a specific

coverage. Since the model is trained to give predictions with an optimal selective risk for the coverage for which they were optimized, the selective risk at that coverage is the most relevant metric by which to measure the performance of a SelectiveNet model. The paper that introduced SelectiveNet [9] emphasized the need to calibrate SelectiveNet's trained coverage to fit the desired *empirical coverage*. For fairness and to reach flawless calibration, we train the models to fit a trained coverage of 0.7, but evaluate the selective risk according to the model's true empirical coverage on the test set (as observed without any attack). For example, the empirical coverage of SelectiveNet with ResNet18 as its backbone on the test set is 0.648, and the selective risk we show in Table 4 is the model's risk for that exact coverage. As additional metrics, we also provide the AURC, NLL and Brier score (even though they do not reflect the true goal of the network in this context). We consider a *direct*

Table 4: Results of ACE on SelectiveNet when used with ResNet18 and VGG16 as a backbone, and trained on CIFAR-10. The empirical coverages (perfectly calibrated) for ResNet18 and VGG16 are 0.648 and 0.778, respectively. The selector is always used as the confidence score. A direct ACE attack targets the selector, and an indirect attack targets the softmax score.

| | $\epsilon$ | Effective $\epsilon$ | Selective Risk | AURC | NLL | Brier Score | Accuracy |
|---|---|---|---|---|---|---|---|
| | 0 | 0 | 0.0012 | 5.04 | 0.204 | 0.083 | 94.83 |
| ResNet18 Backbone | 0.00005 | 0.00005 | 0.0015 | 5.56 | 0.211 | 0.085 | 94.83 |
| Direct (selector head) | 0.0005 | 0.00049 | 0.0059 | 12.16 | 0.283 | 0.102 | 94.83 |
| | 0.005 | 0.00377 | 0.0747 | 132.08 | 0.547 | 0.151 | 94.83 |
| | 0 | 0 | 0.0012 | 5.04 | 0.204 | 0.083 | 94.83 |
| ResNet18 Backbone | 0.00005 | 0.00005 | 0.0015 | 5.5 | 0.215 | 0.087 | 94.83 |
| Indirect (softmax score) | 0.0005 | 0.00049 | 0.0056 | 11.69 | 0.314 | 0.114 | 94.83 |
| | 0.005 | 0.00385 | 0.0796 | 150.07 | 0.648 | 0.162 | 94.83 |
| | 0 | 0 | 0.0067 | 7.46 | 0.259 | 0.105 | 93.54 |
| VGG16 Backbone | 0.00005 | 0.00005 | 0.0075 | 7.87 | 0.267 | 0.107 | 93.54 |
| Direct (selector head) | 0.0005 | 0.00049 | 0.0163 | 12.57 | 0.34 | 0.128 | 93.54 |
| | 0.005 | 0.00402 | 0.0772 | 116.1 | 0.703 | 0.175 | 93.54 |
| | 0 | 0 | 0.0067 | 7.46 | 0.259 | 0.105 | 93.54 |
| VGG16 Backbone | 0.00005 | 0.00005 | 0.0075 | 7.86 | 0.268 | 0.108 | 93.54 |
| Indirect (softmax score) | 0.0005 | 0.00049 | 0.0163 | 12.52 | 0.35 | 0.134 | 93.54 |
| | 0.005 | 0.00404 | 0.0829 | 122.46 | 0.761 | 0.181 | 93.54 |

*attack* on the model's selector head and an *indirect attack* via the model's softmax score. In both cases the final confidence score is measured using the selector head. As seen in the results in Table 4, an indirect attack by softmax seems more harmful than a direct one. For example, for SelectiveNet with a VGG16 backbone, the error over the not "rejected" instances is 0.67%, but it is 7.72% for directly (selector) crafted adversarial instances ($\epsilon = 0.005$) and 8.29% for indirectly (softmax) crafted adversarial instances ($\epsilon = 0.005$). For context, we include in Appendix E ACE results on standard (not selective) architectures trained on CIFAR-10 with the confidence score being the softmax score.

## 6 Concluding Remarks

We have presented a novel attack on uncertainty estimation, capable of causing significant damage with even small values of $\epsilon$ (compared to standard adversarial attacks) and without changing the model's original predictions. This attack severely harms some of the most popular techniques for estimating uncertainty. We can think of several ways in which the attack could be further improved: First is optimizing the perturbations in an iterative process, similar to how *BIM* [17] or *PGD* [21] take FGSM steps iteratively. This may not only increase ACE's potency, but may also fix the artifacts of larger values of $\epsilon$ leading to a lower selective risk than smaller values (observed in Sections 5.1 and 5.2). This might also have the added benefit of using more of the $\epsilon$ resources that were allocated to the attack, thus increasing the effective $\epsilon$. Low ratios of $\frac{\text{Effective } \epsilon}{\epsilon}$ such as observed from our results mean unspent resources for the attack.

The results of this attack also teach us that some methods are more resilient than others. We hope that this work will encourage efforts to examine the resiliency of other and new uncertainty estimation methods, as well as creating defense mechanisms or detection techniques for them. Finally, in the case of none-0/1 loss functions, an attack on uncertainty estimation might need to completely reverse the partial order observed by the model, which would capture the gaps in uncertainty between two correct instances or between two incorrect instances. Such an attack could be used for regression tasks as well, more naturally than our suggested simple conversion of ACE, discussed in Section 4.

## Acknowledgments

This research was partially supported by the Israel Science Foundation, grant No. 710/18.

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
