# Disrupting Deep Uncertainty Estimation Without Harming Accuracy
# Supplementary Material

**Ido Galil**
Technion
idogalil.ig@gmail.com

**Ran El-Yaniv**
Technion,    Deci.AI
rani@cs.technion.ac.il

## A    Evaluation metrics

Let $V_n$ be an independent set of n labeled points. Given a prediction model $f$ and a confidence score function $\kappa$, we define $\Theta$ as a set of all possible values of $\kappa$ in $V_n$, $\Theta \triangleq \{\kappa(x, \hat{y}_f(x)|f) : (x, y) \in V_n\}$, and assume that $\Theta$ contains $n$ unique values. Then the area under the RC curve (AURC) [2] for $\kappa$ is: $\text{AURC}(\kappa, f|V_n) = \frac{1}{n} \sum_{\theta \in \Theta} \hat{r}(f, g_\theta|V_n)$.

While AURC is a useful and well motivated metric, we also report on two other metrics that are often used in the relevant literature. The Negative Log Likelihood (NLL) is defined as $\sum_{y \in Y} -\ln(p_y)$, with $Y$ being the correct labels and $p_y$ being the probability assigned by the model to label $y$.

The Brier score [1] computed over a sample of size $N$ for which there are $R$ possible labels is defined as:

$$\frac{1}{N} \sum_{i=1}^{N} \sum_{j=1}^{R} (f_{ij} - o_{ij})^2.$$

## B    Attacking softmax uncertainty

Code for the algorithm is available at: https://github.com/IdoGalil/ACE

### B.1    Softmax under white-box settings

Table 1 shows the results of using ACE for different values of $\epsilon$. As was the case for black-box settings, for larger values of $\epsilon$, the effective $\epsilon$ is half the size or less, meaning even fewer resources are needed for a very harmful attack. Under attack by ACE with $\epsilon = 0.005$, the AURC is about seven times worse and the NLL and Brier score are about three times worse. Figure 1 includes all the RC curves for ACE under white-box settings.

### B.2    Adversarial robustness via adversarial training

Table 2 presents an evaluation of ACE performance on an EfficientNetB0 that was adversarially trained [3]. We include the results of a non-adversarially trained EfficientNetB0 for comparison. These results suggest there is no significant gain in robustness to ACE by using standard adversarial training. We hypothesize that it is due to adversarial training focusing on crafting examples with a very high loss, or that are able to cross the decision boundary. Such examples require a relatively large $\epsilon$. Adversarial training using ACE or that uses various smaller values of $\epsilon$ might be able to improve robustness to ACE.

35th Conference on Neural Information Processing Systems (NeurIPS 2021).

Table 1: Results of ACE under white-box settings on various architectures pretrained on ImageNet with softmax as the confidence score.

| White-box | $\epsilon$ | Effective $\epsilon$ | AURC | NLL | Brier Score | Accuracy |
|---|---|---|---|---|---|---|
| ResNet50 | 0 | 0 | 69.9 | 0.963 | 0.336 | 76.01 |
| | 0.00005 | 0.000049 | 86.1 | 1.037 | 0.371 | 76.01 |
| | 0.0005 | 0.000422 | 269 | 1.639 | 0.562 | 76.01 |
| | 0.005 | 0.002245 | 555.4 | 3.237 | 0.718 | 76.01 |
| EfficientNetB0 | 0 | 0 | 69.1 | 0.958 | 0.322 | 77.67 |
| | 0.00005 | 0.000049 | 81 | 1 | 0.343 | 77.67 |
| | 0.0005 | 0.000446 | 291.5 | 1.416 | 0.516 | 77.67 |
| | 0.005 | 0.002563 | 553.3 | 2.837 | 0.815 | 77.67 |
| Mobilenet V2 | 0 | 0 | 89.7 | 1.147 | 0.386 | 71.85 |
| | 0.00005 | 0.000049 | 112 | 1.232 | 0.428 | 71.85 |
| | 0.0005 | 0.000397 | 377.3 | 1.973 | 0.671 | 71.85 |
| | 0.005 | 0.001934 | 624.8 | 3.854 | 0.789 | 71.85 |
| DenseNet161 | 0 | 0 | 66.8 | 0.945 | 0.326 | 77.15 |
| | 0.00005 | 0.000049 | 82.6 | 1.02 | 0.36 | 77.15 |
| | 0.0005 | 0.000425 | 261 | 1.643 | 0.54 | 77.15 |
| | 0.005 | 0.002153 | 521.6 | 3.392 | 0.68 | 77.15 |
| VGG16 | 0 | 0 | 80.5 | 1.065 | 0.366 | 73.48 |
| | 0.00005 | 0.000049 | 103.4 | 1.164 | 0.413 | 73.48 |
| | 0.0005 | 0.000387 | 361.7 | 2.015 | 0.646 | 73.48 |
| | 0.005 | 0.001832 | 572.3 | 4 | 0.7 | 73.48 |

Table 2: Comparison of an EfficientNetB0 adversarially trained with a non-adversarially trained EfficientNetB0 in both black-box and white-box settings

| Effects of Adversarial Training | $\epsilon$ | Effective $\epsilon$ | AURC | NLL | Brier Score | Top1 Accuracy |
|---|---|---|---|---|---|---|
| EfficientNetB0 (Black-box) | 0 | 0.00000 | 69.1 | 0.958 | 0.322 | 77.67 |
| | 0.0005 | 0.00049 | 78.8 | 1 | 0.342 | 77.67 |
| | 0.005 | 0.00446 | 185.1 | 1.334 | 0.47 | 77.67 |
| | 0.05 | 0.02902 | 300.9 | 1.775 | 0.585 | 77.67 |
| EfficientNetB0 AdvProp (Black-box) | 0 | 0.00000 | 73.4 | 1.009 | 0.337 | 76.56 |
| | 0.0005 | 0.00050 | 77.9 | 1.029 | 0.346 | 76.56 |
| | 0.005 | 0.00475 | 127.3 | 1.206 | 0.421 | 76.56 |
| | 0.05 | 0.03322 | 295.5 | 1.736 | 0.686 | 76.56 |
| EfficientNetB0 (White-box) | 0 | 0.000000 | 69.1 | 0.958 | 0.322 | 77.67 |
| | 0.00005 | 0.000049 | 81 | 1 | 0.343 | 77.67 |
| | 0.0005 | 0.000446 | 291.5 | 1.416 | 0.516 | 77.67 |
| | 0.005 | 0.002563 | 553.3 | 2.837 | 0.815 | 77.67 |
| EfficientNetB0 AdvProp (White-box) | 0 | 0.000000 | 73.4 | 1.009 | 0.337 | 76.56 |
| | 0.00005 | 0.000050 | 78.5 | 1.028 | 0.346 | 76.56 |
| | 0.0005 | 0.000476 | 144.7 | 1.211 | 0.429 | 76.56 |
| | 0.005 | 0.003172 | 569.9 | 2.537 | 0.79 | 76.56 |

## B.3 Risk-coverage curves for softmax under black-box settings

Figure 2 includes all of the RC curves for ACE under black-box settings. Note that for MobileNetV2, VGG16 and ResNet50, the $\epsilon$ for the most potent attack depends greatly on which coverage is targeted. For example, the selective risk for MobileNetV2 on coverage 0.6 is *higher* for $\epsilon = 0.005$ than it is for $\epsilon = 0.05$.

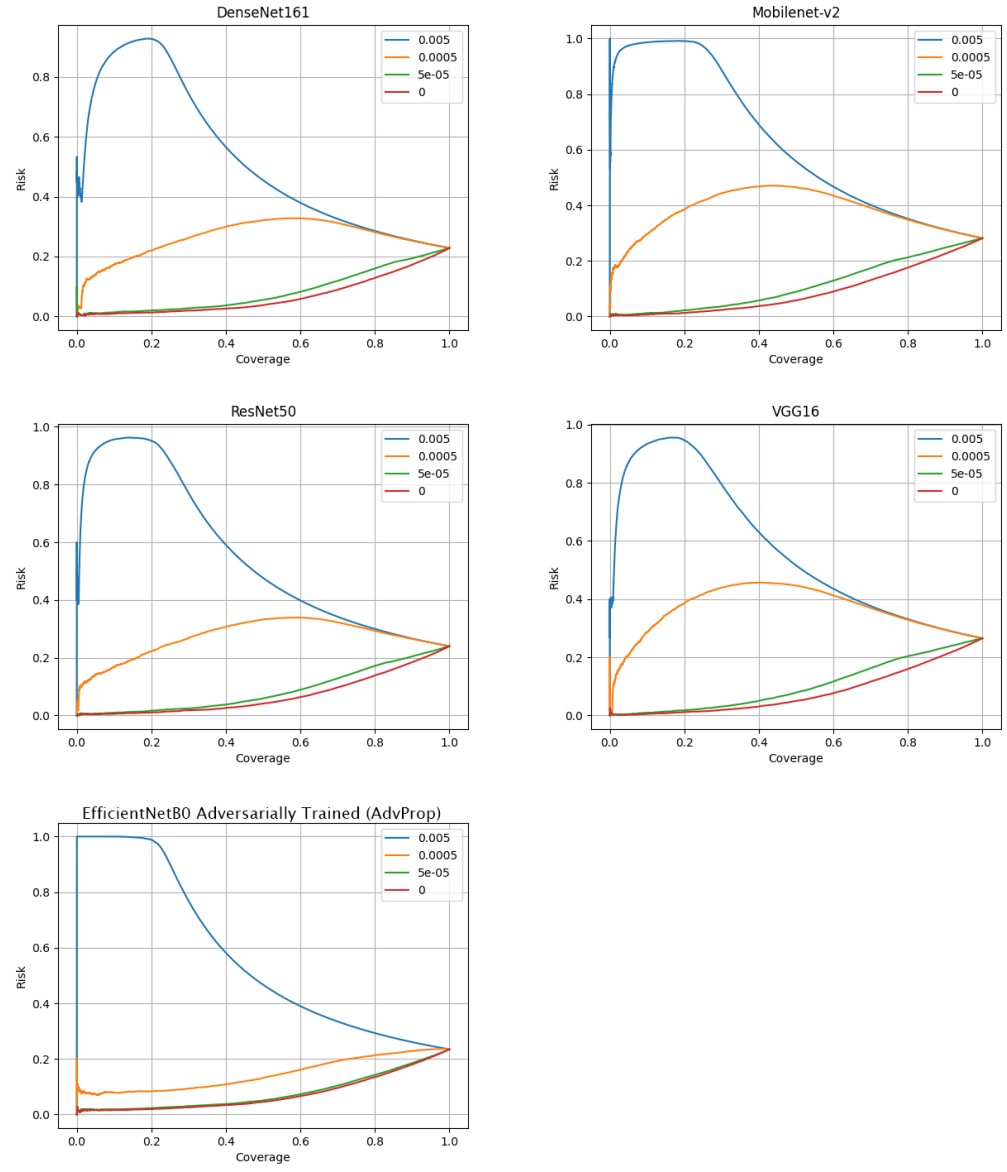

Figure 1: RC Curves of ACE being used under white-box settings with softmax

## C Deep ensembles

### C.1 Deep ensembles under white-box settings

Table 3 shows the results of ACE on ensembles of different sizes consisting of ResNet50 models trained on ImageNet. Under attack by ACE with $\epsilon = 0.005$, the AURC degrades about eightfold. Note that the bigger the ensemble, the more resilient it is to ACE, and even the smallest ensemble is more resilient than a single ResNet50 model (an ACE used on a single ResNet50 model is presented in Table 1). The resulting RC curves are shown in Figure 3.

### C.2 Risk-coverage curve for deep ensembles under black-box settings

Figure 4 includes the RC curve for using ACE on an ensemble of size 5 consisting of ResNet50 models under black-box settings, attacking it with either an equal ensemble consisting of different ResNet50 models or a "foreign" ensemble (consisting of EfficientNet, MobileNet and VGG). Note

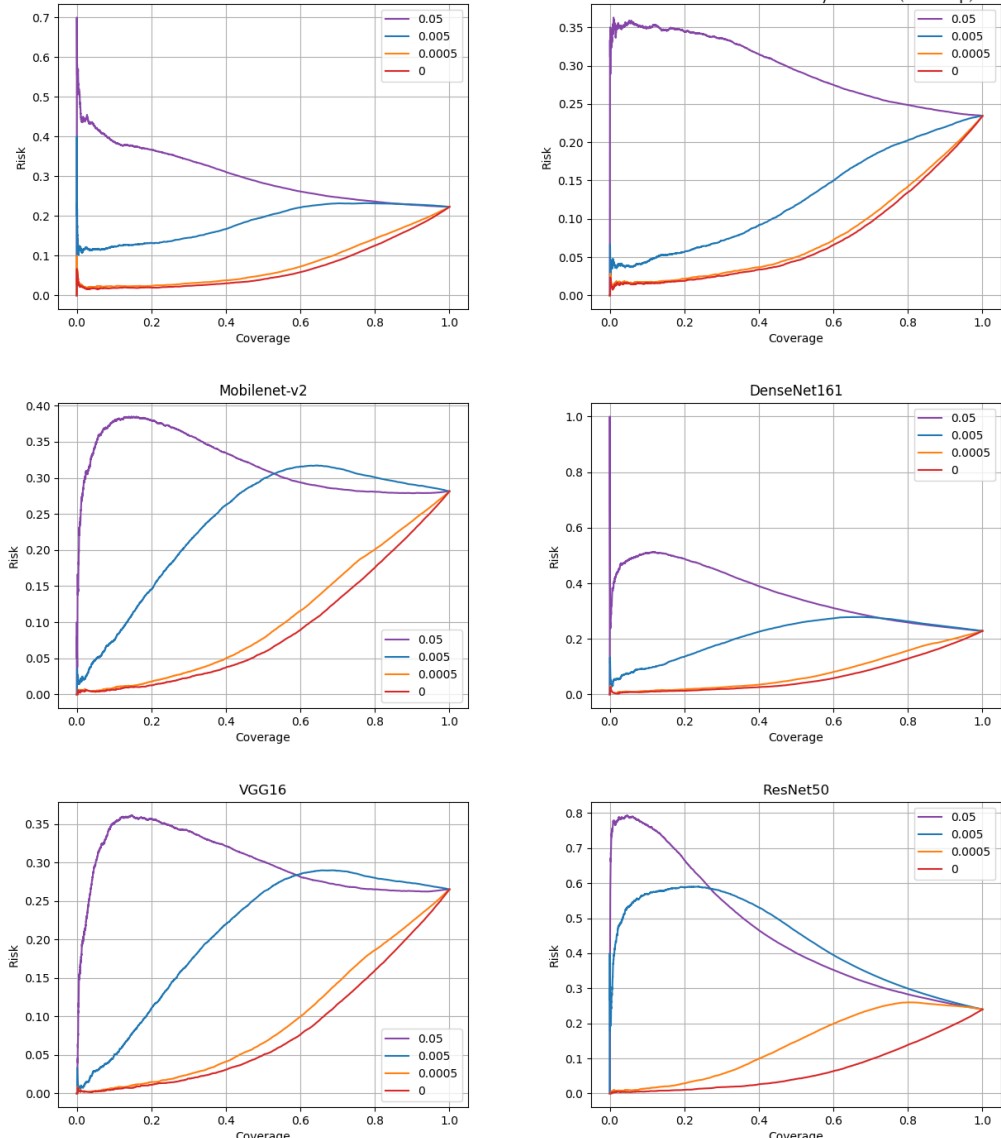

Figure 2: RC Curves of ACE being used under black-box settings with softmax

that for the ResNet50 proxy, the selective risk for any coverage above 0.45 is slightly higher for $\epsilon = 0.005$ than it is for $\epsilon = 0.05$, meaning it is more effective to use a smaller $\epsilon$ for these values. We observe the same to a lesser extent when using a foreign proxy; the selective risk for any coverage above 0.8 is slightly higher for $\epsilon = 0.005$ than it is for $\epsilon = 0.05$.

# D    Monte Carlo dropout

## D.1    Monte Carlo dropout under white-box settings

We use ACE on MC-Dropout under white-box settings, with predictive entropy as its confidence score, using either an indirect attack (targeting softmax) or a direct attack (targeting predictive entropy). The results for the indirect softmax method are shown in Table 4 and Figure 5, and results for the direct predictive entropy method appear in Table 5 and Figure 6.

Table 3: Results of ACE under white-box settings on ensembles of different sizes consisting of ResNet50 models trained on ImageNet.

| | $\epsilon$ | Effective $\epsilon$ | AURC | NLL | Brier Score | Accuracy |
|---|---|---|---|---|---|---|
| ResNet50 Ensemble Size 10 | 0 | 0 | 63 | 0.871 | 0.314 | 77.82 |
| | 0.00005 | 4.97E-05 | 68.8 | 0.897 | 0.327 | 77.82 |
| | 0.0005 | 0.000467573 | 133.7 | 1.112 | 0.425 | 77.82 |
| | 0.005 | 0.003325908 | 510.5 | 2.103 | 0.678 | 77.82 |
| ResNet50 Ensemble Size 5 | 0 | 0 | 63.8 | 0.883 | 0.317 | 77.61 |
| | 0.00005 | 4.96E-05 | 71.2 | 0.916 | 0.333 | 77.61 |
| | 0.0005 | 0.000460713 | 154.8 | 1.177 | 0.449 | 77.61 |
| | 0.005 | 0.003261595 | 523.3 | 2.204 | 0.696 | 77.61 |
| ResNet50 Ensemble Size 3 | 0 | 0 | 65.3 | 0.901 | 0.321 | 77.2 |
| | 0.00005 | 4.95E-05 | 74.4 | 0.94 | 0.34 | 77.2 |
| | 0.0005 | 0.000453732 | 176.9 | 1.252 | 0.474 | 77.2 |
| | 0.005 | 0.003114596 | 533.4 | 2.344 | 0.713 | 77.2 |

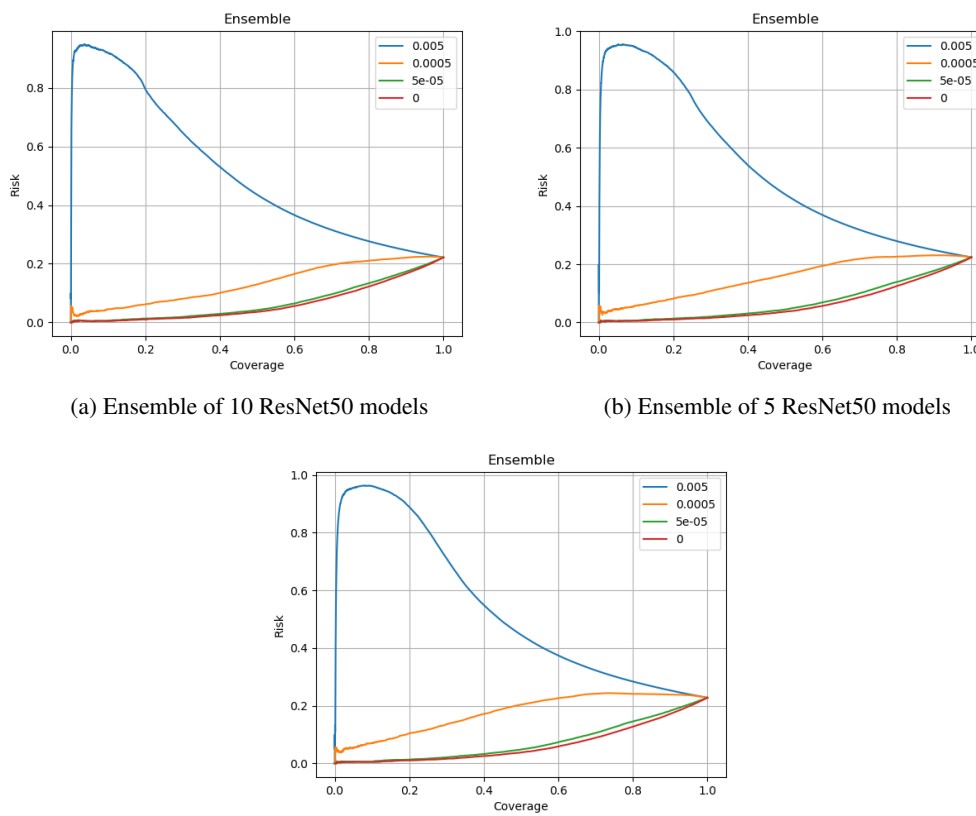

(a) Ensemble of 10 ResNet50 models

(b) Ensemble of 5 ResNet50 models

(c) Ensemble of 3 ResNet50 models

Figure 3: RC Curves resulting from using ACE under white-box settings on ensembles of different sizes consisting of ResNet50 models.

We can observe two interesting phenomena: First, while for small $\epsilon$ values the direct method is slightly more harmful, for bigger values the indirect softmax method is significantly more harmful. Secondly, the more forward passes used for MC-Dropout, the weaker a direct attack becomes. The effects of more forward passes are not as clear for an indirect attack, and they sometimes seem to cause an even stronger attack (as observed in the example of VGG16).

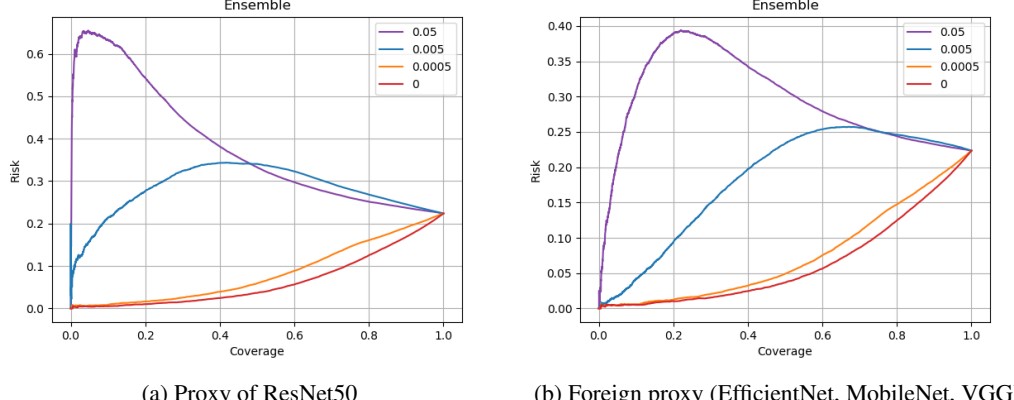

(a) Proxy of ResNet50                    (b) Foreign proxy (EfficientNet, MobileNet, VGG)

Figure 4: RC Curve resulting from using ACE under black-box settings on an ensemble of size 5 consisting of ResNet50 models trained on ImageNet when the proxy is either an equal (but different) ensemble of ResNet50 models, or by a "foreign" ensemble made up of various different architectures (EfficientNet, MobileNet, VGG).

Table 4: Results of ACE on MobileNetV2 and VGG16 pretrained on ImageNet with predictive entropy over several dropout-enabled passes producing the confidence score, and ACE targeting the softmax score of a single dropout-disabled pass (an indirect attack).

| Indirect Attack (Softmax) | $\epsilon$ | Effective $\epsilon$ | AURC | NLL | Brier Score | Accuracy |
|---|---|---|---|---|---|---|
| MobileNet V2 30 passes | 0 | 0 | 94.7 | 1.143 | 0.386 | 71.74 |
| | 0.00005 | 0.000049 | 109.4 | 1.221 | 0.425 | 71.82 |
| | 0.0005 | 0.000397 | 343.2 | 1.914 | 0.669 | 71.76 |
| | 0.005 | 0.001935 | 622.9 | 3.74 | 0.8 | 71.85 |
| MobileNet V2 10 passes | 0 | 0 | 95 | 1.149 | 0.387 | 71.64 |
| | 0.00005 | 0.000049 | 109.8 | 1.224 | 0.424 | 71.69 |
| | 0.0005 | 0.000397 | 342.7 | 1.917 | 0.665 | 71.7 |
| | 0.005 | 0.001946 | 625.6 | 3.767 | 0.799 | 71.62 |
| VGG16 30 passes | 0 | 0 | 86 | 1.057 | 0.366 | 73.32 |
| | 0.00005 | 0.000049 | 100.7 | 1.139 | 0.406 | 73.34 |
| | 0.0005 | 0.000388 | 333.7 | 1.894 | 0.648 | 73.33 |
| | 0.005 | 0.001839 | 502.9 | 3.725 | 0.758 | 73.38 |
| VGG16 10 passes | 0 | 0 | 86.8 | 1.067 | 0.368 | 73.14 |
| | 0.00005 | 0.000048 | 101.5 | 1.146 | 0.406 | 73.18 |
| | 0.0005 | 0.000389 | 332.8 | 1.907 | 0.643 | 73.17 |
| | 0.005 | 0.001849 | 408 | 3.777 | 0.753 | 73.16 |

## D.2 Risk-coverage curves for Monte Carlo dropout under black-box settings

Using the knowledge gained from the white-box experiments, we attack MC-Dropout in black-box settings identically to how we attacked softmax: by using an ensemble as a proxy and attacking its softmax score, without using dropout at all. Observing Figure 7, which shows the RC curves for this setting, we can see that more forward passes made by the models do not seem to affect ACE's effectiveness.

## D.3 Monte Carlo dropout when measuring epistemic uncertainty

We additionally study ACE's impact on MC-Dropout (in white-box settings) when measuring the model's uncertainty (epistemic). We measure it by the variance of the label probabilities. We test ACE

Table 5: Results of ACE on MobileNetV2 and VGG16 pretrained on ImageNet with predictive entropy over several dropout-enabled passes as the confidence score, and targeted by ACE (a direct attack).

| Direct attack (Entropy) | $\epsilon$ | Effective $\epsilon$ | AURC | NLL | Brier Score | Accuracy |
|---|---|---|---|---|---|---|
| MobileNet V2 30 passes | 0 | 0 | 94.4 | 1.143 | 0.386 | 71.78 |
| | 0.00005 | 0.000049 | 110.8 | 1.212 | 0.417 | 71.84 |
| | 0.0005 | 0.000403 | 323.1 | 1.820 | 0.623 | 71.79 |
| | 0.005 | 0.001822 | 587.6 | 3.235 | 0.758 | 71.78 |
| MobileNet V2 10 passes | 0 | 0 | 94.9 | 1.148 | 0.387 | 71.69 |
| | 0.00005 | 0.000048 | 111.3 | 1.215 | 0.417 | 71.71 |
| | 0.0005 | 0.000403 | 321.9 | 1.818 | 0.620 | 71.61 |
| | 0.005 | 0.001826 | 587.9 | 3.244 | 0.754 | 71.73 |
| VGG16 30 passes | 0 | 0 | 85.8 | 1.056 | 0.365 | 73.37 |
| | 0.00005 | 0.000049 | 102.4 | 1.131 | 0.398 | 73.35 |
| | 0.0005 | 0.000392 | 312.8 | 1.782 | 0.606 | 73.31 |
| | 0.005 | 0.001686 | 433.9 | 3.228 | 0.718 | 73.41 |
| VGG16 10 passes | 0 | 0 | 86.7 | 1.067 | 0.368 | 73.17 |
| | 0.00005 | 0.000048 | 102.7 | 1.138 | 0.399 | 73.21 |
| | 0.0005 | 0.000391 | 308.4 | 1.783 | 0.599 | 73.26 |
| | 0.005 | 0.001704 | 521.2 | 3.270 | 0.715 | 73.20 |

in both indirect (targeting softmax) and direct (targeting variance) settings, and use 10 forward-passes for each.

The results are presented in Table 6, with its matching RC curves in Figure 8. These results suggest a few interesting points: (1) epistemic uncertainty is vulnerable to ACE. (2) Similarly to when using preditive entropy, an indirect attack on softmax is more potent than attacking variance directly. (3) We included a large $\epsilon$ (for white-box settings) of $0.05$ to highlight that values too large may lower ACE potency, and that sometimes smaller values of $\epsilon$ are preferable.

# E  CIFAR10 with softmax confidence score

Table 7 shows the results of using ACE on ResNet18 and VGG16 trained on CIFAR-10 with softmax as the confidence score.

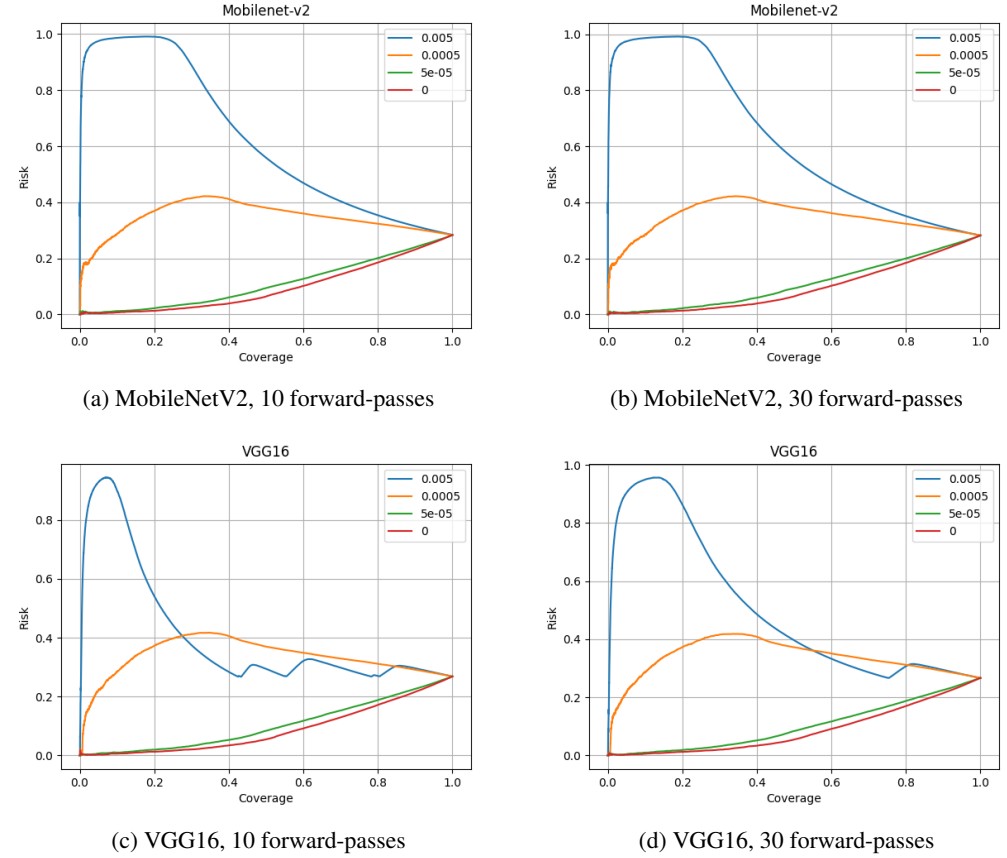

(a) MobileNetV2, 10 forward-passes

(b) MobileNetV2, 30 forward-passes

(c) VGG16, 10 forward-passes

(d) VGG16, 30 forward-passes

Figure 5: RC Curves of an indirect white-box attack by ACE targeting softmax and tested on MC-Dropout using a various amounts of forward-passes.

Table 6: Results of ACE on MobileNetV2 and VGG16 pretrained on ImageNet with variance over several dropout-enabled passes as the confidence score. ACE either targets the variance itself (a direct attack) or softmax (an indirect attack).

| MC-Dropout Variance (Epistemic) | $\epsilon$ | Effective $\epsilon$ | AURC | NLL | Brier Score | Accuracy |
|---|---|---|---|---|---|---|
| MobileNet V2 Indirect (Softmax) | 0 | 0 | 120.4 | 1.15 | 0.387 | 71.61 |
| | 0.0005 | 0.0004 | 370.6 | 1.916 | 0.665 | 71.754 |
| | 0.005 | 0.00194 | 624.3 | 3.769 | 0.799 | 71.646 |
| | 0.05 | 0.01377 | 444.9 | 2.623 | 0.682 | 71.716 |
| MobileNet V2 Direct (Variance) | 0 | 0 | 119.5 | 1.149 | 0.387 | 71.75 |
| | 0.0005 | 0.00037 | 322.3 | 1.479 | 0.501 | 71.76 |
| | 0.005 | 0.00148 | 453.2 | 2.07 | 0.566 | 71.82 |
| | 0.05 | 0.00984 | 414.3 | 1.78 | 0.545 | 71.73 |
| VGG16 Indirect (Softmax) | 0 | 0 | 112.1 | 1.069 | 0.368 | 73.084 |
| | 0.0005 | 0.00039 | 361.6 | 1.904 | 0.643 | 73.06 |
| | 0.005 | 0.00185 | 565.9 | 3.771 | 0.753 | 73.104 |
| | 0.05 | 0.01514 | 448 | 2.642 | 0.679 | 73.114 |
| VGG16 Direct (Variance) | 0 | 0 | 112.4 | 1.069 | 0.368 | 73.08 |
| | 0.0005 | 0.00037 | 297.1 | 1.398 | 0.479 | 73.11 |
| | 0.005 | 0.00142 | 414.5 | 1.932 | 0.537 | 73.07 |
| | 0.05 | 0.01078 | 394.3 | 1.693 | 0.528 | 73.09 |

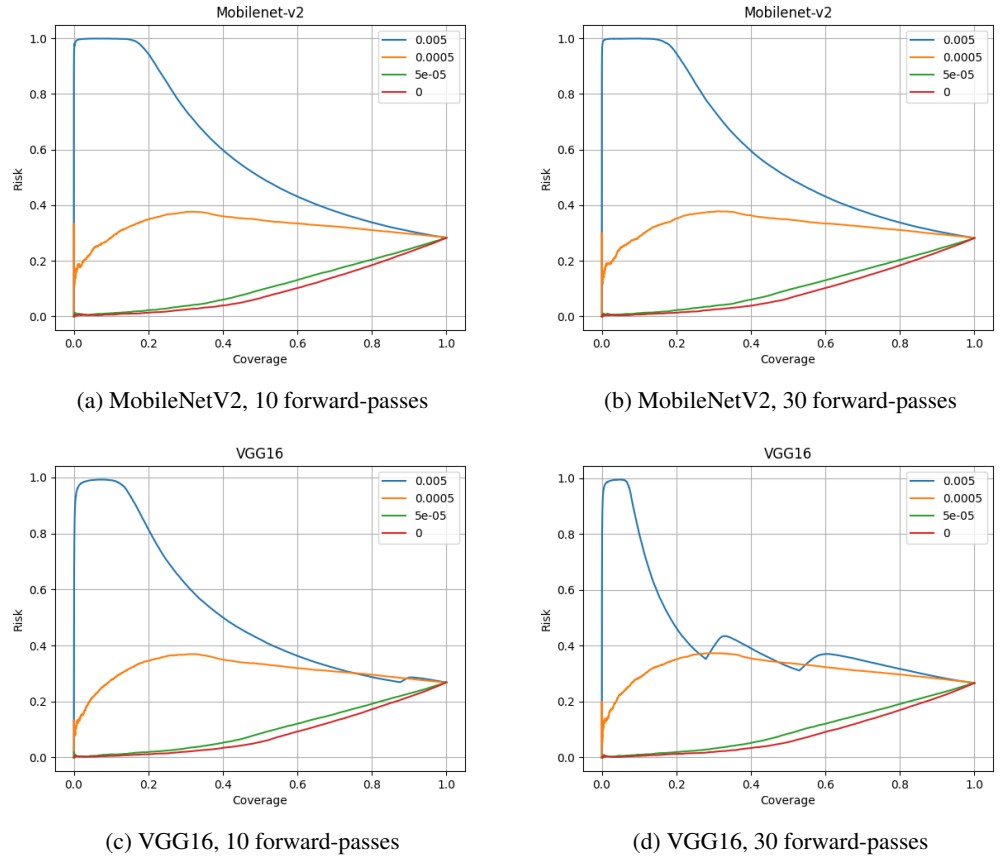

(a) MobileNetV2, 10 forward-passes

(b) MobileNetV2, 30 forward-passes

(c) VGG16, 10 forward-passes

(d) VGG16, 30 forward-passes

Figure 6: RC Curves of a direct white-box attack by ACE targeting the predictive entropy directly and tested on MC-Dropout using various amounts of forward-passes (matching the amount targeted).

Table 7: Results of ACE on ResNet18 and VGG16 trained on CIFAR-10 and softmax as the confidence score.

|  | $\epsilon$ | Effective $\epsilon$ | AURC | NLL | Brier Score | Accuracy |
|---|---|---|---|---|---|---|
| ResNet18 | 0 | 0 | 6.5 | 0.229 | 0.083 | 94.98 |
|  | 0.00005 | 0.00005 | 7.2 | 0.239 | 0.087 | 94.98 |
|  | 0.0005 | 0.00049 | 16.1 | 0.331 | 0.111 | 94.98 |
|  | 0.005 | 0.00385 | 175 | 0.728 | 0.152 | 94.98 |
| VGG16 | 0 | 0 | 8.5 | 0.322 | 0.116 | 93.17 |
|  | 0.00005 | 0.00005 | 9.1 | 0.333 | 0.12 | 93.17 |
|  | 0.0005 | 0.00049 | 15.9 | 0.426 | 0.145 | 93.17 |
|  | 0.005 | 0.00395 | 133.8 | 0.789 | 0.18 | 93.17 |

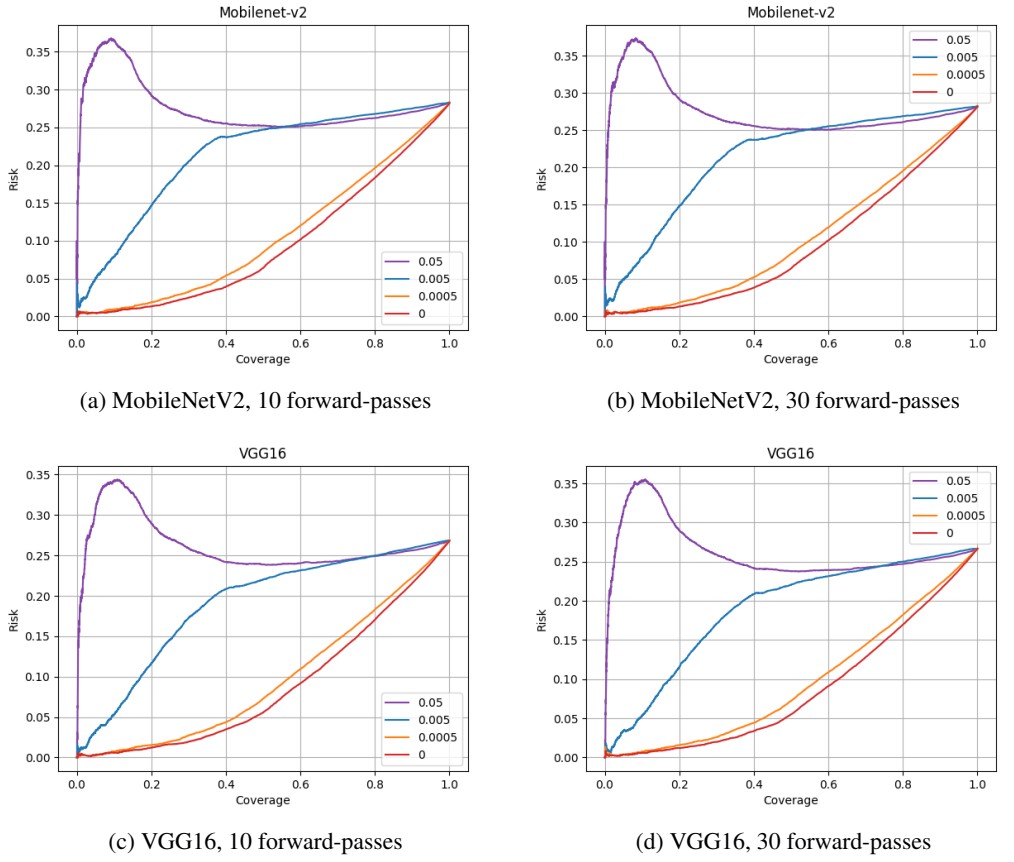

(a) MobileNetV2, 10 forward-passes

(b) MobileNetV2, 30 forward-passes

(c) VGG16, 10 forward-passes

(d) VGG16, 30 forward-passes

Figure 7: RC Curves of a black-box attack by ACE targeting the softmax of the ensemble proxy tested on MC-Dropout using predictive entropy and various amounts of forward-passes.

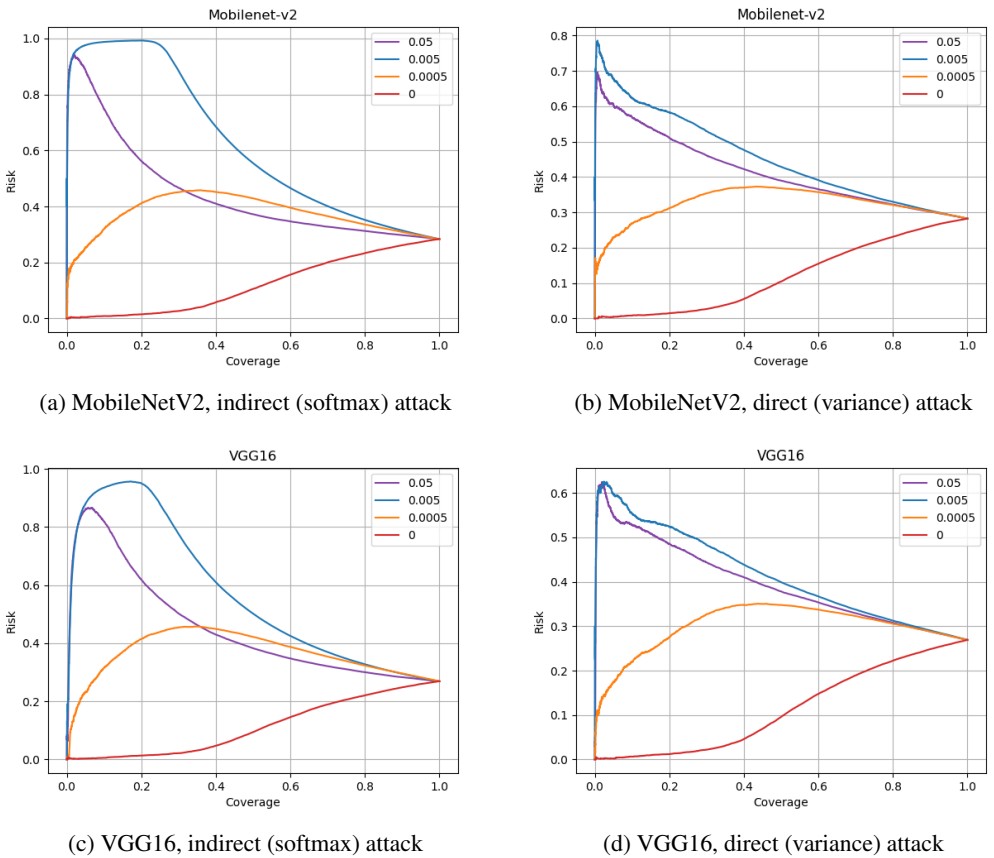

(a) MobileNetV2, indirect (softmax) attack

(b) MobileNetV2, direct (variance) attack

(c) VGG16, indirect (softmax) attack

(d) VGG16, direct (variance) attack

Figure 8: RC Curves of a white-box attack by ACE targeting either the variance of 10 dropout enabled forward-passes for a direct setting, or softmax (with dropout disabled) for an indirect setting.