# OpenReview forum: "Disrupting Deep Uncertainty Estimation Without Harming Accuracy"
_NeurIPS.cc/2021/Conference — NeurIPS 2021 Poster_

### Official Review · Reviewer_XqTW · 2021-07-14

**Rating:** 7
**Confidence:** 4

**Summary:**

The paper presents an interesting method to create adversarial examples that do not change the class label, but rather flips the confidence in that label.  In other words, the accuracy of the classification system is not changed.  Rather, the idea is that the attack will force a system with well calibrated confidence to become confidently wrong and unconfidently correct rather than the desired behavior of being unconfidently wrong and confidently correct. The overall approach is basically the same as a standard label adversarial attack (i.e., follow the gradient), but at a lower magnitude as not to change the label.  The other minor difference is to determine the direction of the gradient depending on the ground truth label of the test sample.  That is increase confidence if the classification label prediction is incorrect and reduce the confidence otherwise. The paper does include extensive experimentation over various neural network models including uncertainty-aware models such as ensembles, MC dropout and SelectiveNet.

**Ethical Concerns:**

No concerns.  The datasets do not seem to divulge any PII and there are not human-subject experiments.

**Limitations And Societal Impact:**

The paper is about possible attacks to uncertainty-aware machine learning classifiers that could reduce human trust in the system while the reported accuracy of the system looks good.  Understanding this vulnerability is the first step of designing machine learning systems that will be robust to such a vulnerability.  There does not seem to be any negative societal impact of this work.

**Main Review:**

At one level, the proposed method seems to be a corollary of creation of adversarial examples, i.e., standard attack at low magnitude to mess with confidence but not change the label. The slight innovation is to use knowledge of the ground truth and predicted labels to determine which direction to follow the gradient.   This leads to my main concern of the paper.  It requires knowledge of the ground truth.  To create one adversarial example, this might not be a problem as one may argue that through costly manual intervention an attacker can determine the ground truth.  However, for this method to be effective in terms of reducing human trust in the classifier label recommendations, the attack has to be executed at scale so that the confidence is flipped for most test samples.  Now knowledge of the ground truth by the attacker at scale seems too costly. Perhaps an attacker can simply flip the reported confidence without knowing the ground truth. It would be interesting to evaluate the efficacy of such a more realistic method.

The experiments do provide very interesting insights including the efficacy of choosing epsilon and the resiliency of ensemble methods.

The experiments focus on aleatoric uncertainty for the ensemble and MC dropout methods in focusing on softmax and entropy.  In other words, the focus in on the posterior predictive (or sample mean of the label probabilities).  It would be interesting to understand how the attacks work on epistemic uncertainty which could be measured as the spread or variance of the label probabilities.

There seems to be a discrepancy between the AURC numbers in Table 1 and Figure 4 for the EfficientNetB0 model.

In light of the authors' response that the more practical attacks will be discussed and the paradigm shift the paper offers in terms of adversarial examples, I can recommend acceptance of the paper.

**Time Spent Reviewing:**

3

---

> ### Author Response · Authors · 2021-08-06
> **Scalability, potential improvements and epistemic uncertainty**
>
> Your concerns about the feasibility of an attacker gaining ground truths at a large scale are valid. However, some attackers could have a good proxy to the ground truths, such as a different model capable of making good predictions. Alternatively, the attacker might not care at all about the ground truths and would like only to increase the confidence for certain types of labels and decrease the confidence for other types regardless of their truthfulness (i.e. causing a loan company to have a very high confidence for the label approving giving a loan and a very low confidence for the label disapproving loans, increasing the chance for it to go bankrupt).
> As you have suggested (and we've also briefly mentioned in Sections 4 and 6), a more refined attack that reverses the partial order induced by the model is sensible. This approach can solve the scalability concern since it will turn the model's uncertainty estimation against itself. We will emphasize this issue in the paper.
>
> We liked your suggestion to evaluate the attack on epistemic uncertainty. In fact, we conducted some experiments with MC-Dropout where we set the uncertainty estimation metric to be variance rather than entropy. Preliminary results suggest it is as vulnerable to ACE as entropy. More interestingly, it seems that using more MC-Dropout forward passes for this variance-based uncertainty estimation lead to an improvement in ACE performance. For example, attacking VGG-16 with variance using 10 dropout iterations results in 387 AURC, whereas 20 dropout iterations results in 404 AURC (which is much worse). This trend is consistent across architectures and epsilons.
> We will add these new results to the paper.
>
> *"There seems to be a discrepancy between the AURC numbers in Table 1 and Figure 4 for the EfficientNetB0 model"*:
>
> Sorry for the misunderstanding. Table 1 shows results for the *black-box* settings, for which the RC curve is presented in Appendix C.2, Figure 3.
> Figure 4 relates to EfficientNetB0 under *white-box* settings whose corresponding table with all the results is in Appendix C.1 in Table 1.
> While this is already mentioned in the caption, we will emphasize it in the revised paper.

---

> > ### Comment · Reviewer_XqTW · 2021-08-19
> > **Response to Scalability, potential improvements and epistemic uncertainty**
> >
> > The authors have addressed my concerns.
> >
> > As I also read the response to reviewer qDh3, I realize that there are ways other than simple monitoring the accuracy that one can detect the ACE attack. For instance one could over time compute AURC values for the confidence ($\max_i p_i$) and for its reciprocal (reverse ordering).  If the reciprocal leads to much lower AURC, then there is an ACE attack.
> >
> > The idea of the ACE is very simple and it piggybacks on current adversarial methods.  On the other hand, the ides is novel and provides a fresh perspective at how to conduct an attack. In this light, I can raise my recommendation to an accept given that the authors add discussion about more practical and scalable attacks and about how new detection methods that go beyond monitoring accuracy could be formulated.
> >
> > Certainly, the results on variance as an uncertainty measure would also be interesting.

---

> > > ### Author Response · Authors · 2021-08-20
> > > **Detection of ACE**
> > >
> > > Thanks for your positive and constructive feedback.
> > > We will of course add such a discussion about more practical and scalable attacks, along with a separate discussion of how to detect such attacks.
> > >
> > > In regards to detection. Your suggestion is interesting and will obstruct the standard ACE. A variation of ACE adapting to such a detection mechanism could be the following: rather than reversing the partial order, it could instead only render it meaningless (i.e. by only increasing the confidence for incorrect samples), so trying to flip the values will result in having the same AURC. Although the potency of ACE will decrease, the selective risk will still be greatly damaged and equal to the model's overall error rate.
> > > Another variation could be to use ACE "surgically", targeting only few but very influential samples. For example, in the case of a loan company it could only raise the relative confidence (ranking) of risky applicants that requested the highest loans. Such a small-scale attack will have little effect on the value of AURC but a significant impact for the company's revenue. Such surgical attacks are extremely hard to detect due to the small number of affected samples and the small epsilon the attack requires.
> > > We will discuss all of these as well as other approaches for defense in the revised paper.

---

> > > > ### Comment · Reviewer_XqTW · 2021-08-24
> > > > **Response to Detection of ACE**
> > > >
> > > > Yes, the arms race continues as you proposed a counter-counter-attack to my  suggested counter-attack.  The main point is that the paper provides a fresh perspective and paradigm shift about how to disrupt machine learning applications with adversarial examples.

---

### Official Review · Reviewer_qDh3 · 2021-07-17

**Rating:** 6
**Confidence:** 4

**Summary:**

This paper proposes an adversarial attack on uncertainty estimation that does not change the predicted label of a classifier. Rather than perturbing an image to flip its predicted label, this method moves incorrectly classified images further away from the decision boundary (with respect to some notion of distance that depends on the uncertainty method being attacked), thus artificially reducing the model's uncertainty in these images.

**Main Review:**

The presented idea is clearly presented and properly contextualized, and the experimental analysis of the attack is thorough and considers a wide variety of uncertainty estimation algorithms, showing that they are all vulnerable to the proposed attack. I am not an expert on uncertainty estimation, but from my experience both the paper and the evaluation are technically sound.

My primary concern is with the novelty and technical depth of the approach, combined with a lack of showcased "real-world" applications. In particular, technique-wise, the method is just performing weak adversarial attacks (FGSM-based with small epsilon, which are known to preserve the predicted class) that are untargeted on the correctly classified images, and targeted (towards the predicted class) on the incorrectly classified images. Such attacks have been thoroughly studied in the adversarial examples literature, see e.g. https://arxiv.org/abs/1608.04644 or https://arxiv.org/abs/1902.06705, and in general given any "reasonable" (differentiable) loss function derived from the output of an image classifier, it's not really surprising that images can be imperceptibly perturbed to maximize or minimize it.

For this reason, I think that the majority of the novelty here is in the concept/idea of attacking uncertainty estimation, rather in the technical approach. This is (in my opinion) completely fine, but in the absence of a fundamentally new technique, I think that the paper should demonstrate more convincing "real-world" applications of their proposed attacks, rather than focusing on the controlled settings shown in the current evaluation section. For example: are there deployed systems that take action based on uncertainty estimates? If so, does the proposed attack cause misbehavior in these systems?

More specific/detailed comments:
- [Minor] I think "mislabeled instance" in Figure 1 actually be "misclassified instance"
- [Minor] Although it is commonly accepted/well-known, I think along with Figure 2 the paper should provide a quantitative summary (e.g., a simple confidence histogram) illustrating that indeed, on unaltered images, classifiers are more confidence when predicting the correct label than the incorrect label.
- The related work section is rather light on literature relating to adversarial attacks, of which a variety have been proposed, many with the explicit goal of maximizing or minimizing distance to the decision boundary (e.g., the CW attacks, MMA attack, etc.). These attacks and their relation to the proposed methods should be discussed in more depth.
- The definition of RC and AURC curves should not be entirely relegated to the Appendix, since it is central to the evaluation of the proposed method. Appendix A is probably too detailed to put into the paper as-is, but the paper should stand alone and thus would benefit from a short paragraph explaining RC, AURC, and their relevance to evaluating uncertainty estimation.
- [Minor] Based on the motivation section, I thought that the goal was to find the *smallest* epsilon the would ruin the image ordering by confidence, but the algorithm actually searches for the largest epsilon that does not flip the decision boundary. What is the motivation for choosing epsilon in this way (i.e., searching downwards and not upwards)?



**Time Spent Reviewing:**

3

---

> ### Author Response · Authors · 2021-08-06
> **Practicality and technical novelty**
>
> Thank you for your detailed review and helpful suggestions,
>
> *"the paper should demonstrate more convincing "real-world" applications of their proposed attacks"*:
>
> We should indeed elaborate on the practicality and advantage of our proposed attack.
> Let's consider a scenario of a debt landing company using Machine Learning to decide whether to approve or disapprove loans to its clients. An interesting example is:
> https://lendbuzz.com/
> Such a company operates by granting loans to highly probable non-defaulting applicants achieving the highest confidence according to their predictive model. The choice of rejection threshold must depend on uncertainty estimation (in accordance with the company's risk tolerance).
> A malicious attacker might attempt to influence this company to grant loans to the maximal amount of defaulting applicants to increase its odds to go bankrupt. Aware of this risk, the company tries to defend itself in various ways. Two obvious options are:
> 1. Monitor their model's accuracy. if it drops significantly it might point to an adversarial attack. Since normal adversarial attacks harms accuracy significantly, it will alarm such monitors.
> 2. Subscribe to AI firewall services like those given by professional parties such as https://www.robustintelligence.com/ .
> Since standard adversarial attacks generate instances that cross the decision boundary, they require a relatively large epsilon that is easier to detect relative to those required by ACE.
>
> Even if the standard adversarial attack bypassed detection, the crossing adversarial instance they generated has a very high uncertainty estimate due to its proximity to the decision boundary.
>
> ACE, on the other hand, is more harmful in this context because (1) it typically uses a considerably smaller epsilon and is harder to detect; (2) accuracy monitoring is no longer relevant (since ACE does not change accuracy); and (3) ACE will give higher confidence to riskier applicants.
>
> *"My primary concern is with the novelty and technical depth of the approach"*:
>
> While we agree that our proposed technique is simple, we think it is an important first step introducing a complex setting of attacking uncertainty estimation. For example, we briefly suggested (in Sections 4 and 6) a more refined attack for future work that reverses the partial order induced by the attacked model. Such an attack would not only have to deal with independent instances, but will also have to consider the entire ordering of all instances.
>
> *"[Minor] I think "mislabeled instance" in Figure 1 actually be "misclassified instance""*:
>
> Accepted, will be fixed.
>
> *"Although it is commonly accepted/well-known, I think along with Figure 2 the paper should provide a quantitative summary (e.g., a simple confidence histogram) illustrating that indeed, on unaltered images, classifiers are more confidence when predicting the correct label than the incorrect label."*:
>
> We will add this histogram to our revised paper.
>
> *"The related work section is rather light on literature relating to adversarial attacks, of which a variety have been proposed, many with the explicit goal of maximizing or minimizing distance to the decision boundary (e.g., the CW attacks, MMA attack, etc.). These attacks and their relation to the proposed methods should be discussed in more depth."*:
>
> Thanks. All these references are indeed relevant and we will include them in our revision.
>
> *"The definition of RC and AURC curves should not be entirely relegated to the Appendix, since it is central to the evaluation of the proposed method. Appendix A is probably too detailed to put into the paper as-is, but the paper should stand alone and thus would benefit from a short paragraph explaining RC, AURC, and their relevance to evaluating uncertainty estimation."*:
>
> We agree and will include a shorter summary of their definitions in the main paper body.
>
> *"[Minor] Based on the motivation section, I thought that the goal was to find the smallest epsilon the would ruin the image ordering by confidence, but the algorithm actually searches for the largest epsilon that does not flip the decision boundary. What is the motivation for choosing epsilon in this way (i.e., searching downwards and not upwards)?"*:
>
> The attacker sets an epsilon threshold for the algorithm to limit detection risk. The algorithm then tries to maximize the epsilon satisfying the constraint of not crossing the decision boundary. The larger the epsilon, the more confidence values will be pushed further apart.

---

> > ### Comment · Reviewer_qDh3 · 2021-08-31
> > **Re: Response**
> >
> > Thanks to the authors for responding to all of the raised concerns.
> >
> > I have raised my score by 1, since the authors have addressed most of my raised points, and the credit lending illustration illustrates a practical scenario in which this attack may be useful.
> >
> > My main concerns remain similar, however:
> >
> > 1. I don't yet see how this significantly differs from just an adversarial attack but on a slightly modified loss function---i.e., instead of hurting the accuracy of a classifier, the attack just targets a different performance metric (namely, the calibration). Even beyond any concerns about novelty (I think that the change of perspective given by the paper is novel enough), I don't yet see how these attacks are less detectable than adversarial examples by just monitoring the metric they are actually attacking.
> >
> > In the credit scenario, for example, the way to detect an adversarial attack was to monitor accuracy, i.e., how many people did the company give a loan to who then defaulted. I'm not sure why uncertainty attacks would not be detected if the company just monitored the size of the loans given vs the clients' trustworthiness (similar to calibration in NNs).
> >
> > More broadly, I would expect both adversarial attacks and uncertainty attacks to immediately cause the company to lose a lot of money very quickly. In this case, the central claim becomes about the size of epsilon needed to fool practical systems (i.e., maybe uncertainty attacks are harder to defend against). This is a reasonable argument, but needs additional justification over what is presented in the paper, since it is really a claim about how the attack will perform in the wild. For example, additional questions that immediately arise are:
> >
> > - Do the small-epsilon attacks succeed against models trained to be locally smooth, such as adversarially trained networks?
> > - Do the small-epsilon attacks transfer between different model architectures?
> > - Can the small-epsilon attacks be executed even in the black-box setting?
> >
> > 2. While the straightforward nature of the technique is not a bad thing (on the contrary, I appreciate the simplicity), in my opinion the paper must go beyond proposing hypothetical scenarios where the attack could be useful and should do more to convince the reader that this attack will be useful in practical settings.

---

> > > ### Author Response · Authors · 2021-09-02
> > > **Small-epsilons and resiliency to ACE**
> > >
> > > Thanks for your thoughtful comment,
> > >
> > > *"Do the small-epsilon attacks succeed against models trained to be locally smooth, such as adversarially trained networks?"*
> > >
> > > This is an excellent question. By now, and in accordance with your question, we have conducted additional experiments on adversarially trained networks, which indicate that standard adversarial training doesn't improve resiliency to ACE. We include below the results for a black-box ACE setting against an EfficientNetB0 trained with AdvProp (we've included ACE against a non-adversarially trained EfficientNetB0 as well for comparison):
> > >
> > > |                        | $\epsilon$ | Effective $\epsilon$ | AURC  |
> > > |------------------------|------------|----------------------|-------|
> > > | EfficientNetB0 AdvProp | 0          | 0.00000              | 73.7  |
> > > |                        | 0.0005     | 0.00050              | 78.2  |
> > > |                        | 0.005      | 0.00474              | 127.8 |
> > > |                        | 0.05       | 0.03304              | 292.7 |
> > >
> > > |                | $\epsilon$ | Effective $\epsilon$ | AURC  |
> > > |----------------|------------|----------------------|-------|
> > > | EfficientNetB0 | 0          | 0                    | 69.1  |
> > > |                | 0.0005     | 0.00049487           | 78.8  |
> > > |                | 0.005      | 0.004458629          | 185.1 |
> > > |                | 0.05       | 0.02902293           | 300.9 |
> > >
> > > Notice that the AURC values for each epsilon are very similar to the AURC values obtained without adversarial training. These results will be added to the paper.
> > >
> > > *"Do the small-epsilon attacks transfer between different model architectures?"*
> > >
> > > *"Can the small-epsilon attacks be executed even in the black-box setting?"*
> > >
> > > If we understand your questions correctly, the answer to both of these questions is yes: we have conducted several experiments in black-box settings, attacking various architectures and included small epsilons in the range we evaluated the attack with. These results are included in the paper in Table 1, Table 2 and Table 3. These attacks were conducted by using a proxy (consisting of ResNet50 models) unrelated to the victim's model, and crafting transferable examples to be used on the victim's model. So the damage in AURC made to EfficientNetB0 for example is a result of a successful transfer.

---

> > > > ### Comment · Reviewer_qDh3 · 2021-09-03
> > > > **Re: Response**
> > > >
> > > > I appreciate the additional experiments run by the authors and would recommend their inclusion in the paper. I also want to thank the authors for engaging with the comments in the initial review.
> > > >
> > > > I still would have liked to see:
> > > >
> > > > (a) more explicit illustration (not just theoretically) of the real-world applicability of the attack,
> > > >
> > > > (b) more clear presentation (esp. with respect to my initial comments about defining AURC in the main body).
> > > >
> > > > Nevertheless I have increased my score by one since I think the new experiments in combination with the existing results are interesting and are sufficient for publication.

---

### Official Review · Reviewer_bohc · 2021-07-18

**Rating:** 7
**Confidence:** 4

**Summary:**

The paper presents a novel way for adversarial attacks.  This work studies the problem of attacking networks to change their confidence without affecting the accuracy. Such attacks increase the confidence of the network on wrong predictions and decrease the confidence of the networks on correct predictions. Since the accuracy of the network is not the target, this enables the. attacker to use smaller attacks that might not be easily detected. The paper studies both white-box and black-box attacks on different kinds of uncertainty estimation methods including MC-dropout, SelectiveNets, and Ensemble methods. Finally, the paper presents extensive empirical evaluations that support the proposed method.

**Limitations And Societal Impact:**

I do not see concerns regarding the societal impacts. This kind of work is important to develop more robust models.

**Main Review:**

- originality: The problem is novel and different from the works that have studied adversarial attacks which aim to drive the network to generate wrong predictions.
- quality: The authors conducted thorough experiments and evaluated different attacks and uncertainty methods.
- Clarity: The paper is well written and easy to follow.
- Significance: This work is important as the effect of attacks on network confidence can affect the practical deployment of deep learning models especially in critical applications.

**Time Spent Reviewing:**

6

---

### Decision · Program_Chairs · 2021-09-28

**Decision:**

Accept (Poster)

**Comment:**

The paper presents a novel approach that has been very well received by the reviewers. Also, the main concerns that were raised after the initial review have been addressed in the rebuttal and there is consensus in accepting the work.
I personally find the proposed method very interesting as it deals with uncertainty estimation and does so in an elegant way.

**Consistency Experiment:**

NeurIPS has a long history of experimentation. In 2014, NeurIPS ran an experiment in which 10% of submissions were reviewed by two independent committees to quantify the randomness in the review process. This year, we repeated a variant of this experiment to see how the quality of the review process has changed over time.  This paper was part of the experiment and was therefore assigned to two committees (consisting of reviewers, an Area Chair, and a Senior Area Chair) that reached independent decisions.  If both committees made the same recommendation, this recommendation was followed. If a single committee recommended acceptance, the paper was accepted (with the exception of a few cases in which the other committee identified what we considered a fatal flaw, e.g., an error in a key result).

This copy’s committee reached the following decision: **Accept (Spotlight)**

The other committee assigned to the paper recommended **Reject**.  You can find the other set of reviews, along with any follow up discussion with the authors here:
https://openreview.net/forum?id=tgdoUMqlwMv